# CD40L protects against mouse hepatitis virus-induced neuroinflammatory demyelination

**Fareeha Saadi**[1], **Debanjana Chakravarty**[1], **Saurav Kumar**[1], **Mithila Kamble**[1], **Bhaskar Saha**[2], **Kenneth S. Shindler**[3,4], **Jayasri Das Sarma**[1,3]*

**1** Department of Biological Sciences, Indian Institute of Science Education and Research Kolkata, Mohanpur, India, **2** National Centre for Cell Science, Ganeshkhind, Pune, India, **3** Departments of Ophthalmology and, **4** Neurology University of Pennsylvania Scheie Eye Institute, Philadelphia, Pennsylvania, United States of America

☯ These authors contributed equally to this work.
* dassarmaj@iiserkol.ac.in

## Abstract

Neurotropic mouse hepatitis virus (MHV-A59/RSA59) infection in mice induces acute neuroinflammation due to direct neural cell dystrophy, which proceeds with demyelination with or without axonal loss, the pathological hallmarks of human neurological disease, Multiple sclerosis (MS). Recent studies in the RSA59-induced neuroinflammation model of MS showed a protective role of CNS-infiltrating CD4$^+$ T cells compared to their pathogenic role in the autoimmune model. The current study further investigated the molecular nexus between CD4$^+$ T cell-expressed CD40Ligand and microglia/macrophage-expressed CD40 using CD40L$^{-/-}$ mice. Results demonstrate CD40L expression in the CNS is modulated upon RSA59 infection. We show evidence that CD40L$^{-/-}$ mice are more susceptible to RSA59 induced disease due to reduced microglia/macrophage activation and significantly dampened effector CD4$^+$ T recruitment to the CNS on day 10 p.i. Additionally, CD40L$^{-/-}$ mice exhibited severe demyelination mediated by phagocytic microglia/macrophages, axonal loss, and persistent poliomyelitis during chronic infection, indicating CD40-CD40L as host-protective against RSA59-induced demyelination. This suggests a novel target in designing prophylaxis for virus-induced demyelination and axonal degeneration, in contrast to immunosuppression which holds only for autoimmune mechanisms of inflammatory demyelination.

## Author summary

MS is primarily considered an autoimmune CNS disease, but its potential viral etiology cannot be ignored. Myelin-specific CD40L$^+$CD4$^+$ T cells migration into the CNS and resultant neuroinflammation is considered pathogenic in autoimmune MS. In contrast, CD40L$^+$CD4$^+$ T infiltration into the MHV-induced inflamed CNS and their interaction with CD40$^+$ microglia/macrophages are shown to be protective in our study. Considering differential etiology but comparable demyelination and axonal loss, immunosuppressive treatments may not necessarily ameliorate MS in all patients. MHV-induced

**Data Availability Statement:** All relevant data are within the manuscript and its Supporting Information files.

**Funding:** This work was supported by a Department of Biotechnology (DBT), India, research grant (BT/PR 20922/MED/122/37/2016). JDS received this grant. The Council of Scientific and Industrial Research (CSIR) India provided fellowships to F.S. and S.K.; the Ministry of Human Resource Development (MHRD), India provided fellowship to D.C; and the University Grants Commission (UGC), India, provided fellowship to M.K. The funders had no role in study design, data collection and analysis, decision to publish, or preparation of the manuscript.

**Competing interests:** The authors have declared that no competing interests exist.

demyelination in this study indicates that the interaction between CD40L on CD4$^+$T cells and CD40 on microglia/macrophage plays an important protective role against MHV-induced chronic progressive demyelination.

## Introduction

A neurotropic strain of Mouse Hepatitis Virus, MHV-A59, or an isogenic recombinant strain RSA59 infection-induced neuroinflammatory demyelination in an experimental model typically initiates with the activation of CNS-resident neuroglial cells[1]. A schematic representation (Fig 1A) depicts the pathogenesis of intracranial RSA59 infection of 4-week old C57BL/6 mice along with the conventional kinetics of viral RNA, infectious virus particles, and the course of RSA59 induced acute neuroinflammation and chronic progressive demyelination[2]. Days 1–7 p.i. are characterized by an onset of neuroinflammation predominated by innate-immune responses. This is followed by a switch to adaptive immunity where acute inflammation resolves, viral titer goes below the detection limit (by routine plaque assay), and viral RNA persists at very low abundance (days 7–15 p.i.). The gradual increase in myelin loss with or without axonal loss starts as early as day 7 p.i., as observed, and reaches its peak at the chronic phase of neuroinflammation (day 30 p.i.) (Fig 1A) [3–7]. This neurodegeneration mimics certain pathological features of the human neurological disease multiple sclerosis (MS)[4]. Early acute immune inflammation is regulated by innate immune responders, microglia, and astrocytes, as they release specific cytokines/chemokines and trophic factors.

However, glial cells cannot restore homeostasis on their own and depend on the consequent infiltration of peripheral leucocytes, especially CD4$^+$ T cells, that help in the efficient transition from an innate immune response to the adaptive immune response required for viral clearance and resolution of acute neuroinflammation[8]. Growing evidence implies existing cellular cross-talk between resident glial cells and migratory peripheral CD4$^+$ T cells [8–12]. Contributing towards the understanding of the interaction between CD4$^+$ T cells and microglia/macrophages in MHV induced inflammation, we recently showed that the absence of CD4$^+$ T cells resulted in fewer CD11b$^+$ microglia/macrophages in the CNS, even though the number of overall CD45$^+$ inflammatory cells inducing encephalitis was unaltered[9]. Also, the absence of CD4$^+$ T cells in the chronic phase of infection deterred CNS recovery and aggravated RSA59 induced axonal blebbing and demyelination, along with a significantly high accumulation of CD11b$^+$ M2 type microglia/macrophages in the demyelinating plaques[9]. In another study, we also demonstrated significantly high mortality of Ifit2$^{-/-}$ mice due to reduced CD4$^+$ T cells infiltration and reduced activation of microglia/macrophages in the brains [2]. These studies indicated that the absence of CD4$^+$ T cells or their hindered recruitment to the CNS affected microglia/macrophage activation and impeded the inflammatory equilibrium in the CNS, preventing re-establishment of homeostasis at the chronic phase of neuroinflammation. Microglia/macrophages can also regulate CD4$^+$ T cells activation in MHV-JHM infected mice [13]. This bi-directional regulation of CD4$^+$ T cells-microglia/macrophages is fundamental to viral clearance and is a two-step process that can be facilitated by one of the co-stimulatory receptor-ligand pairs CD40-CD40L [14]. In the current study, molecular interaction between CD4$^+$ T cells and microglia was studied using CD40L deletion in the background of demyelination susceptible C57BL/6 mice (CD40L$^{-/-}$) and was compared with wild type C57BL/6 mice (WT).

CD40L is a type II tumor necrosis factor receptor (TNFR) superfamily transmembrane protein with a membrane-bound and a soluble form expressed principally by activated T cells and under severe inflammatory situations on natural killer cells and mast cells [15]. CD40L is a

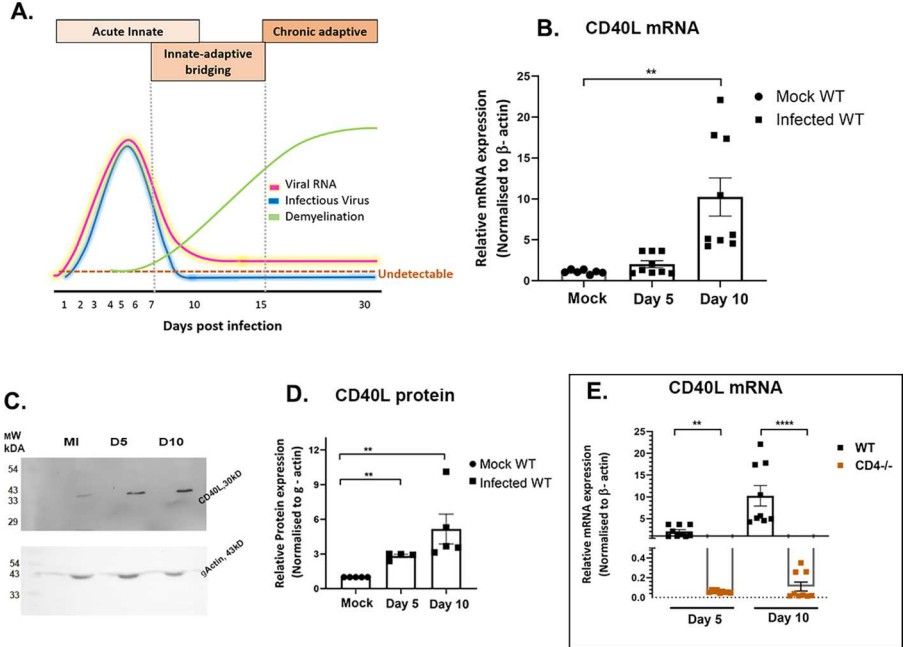

**Fig 1. RSA59 infection alters the expression of CD40L in WT and CD4$^{-/-}$ mouse brains.** (A) Schematic diagram representing the kinetics of MHV-A59/RSA59 viral RNA, infectious particles, and phases of immune response correlating with acute phase neuroinflammation, the bridging of innate and adaptive immune responses, and chronic phase peak of demyelination following intracranial inoculation. Scale-arbitrary. (B) CD40L transcript abundance was determined in whole brain lysates from RSA59 infected (25000 PFUs) WT mice on days 5 and 10 p.i. by qRT-PCR. Results were normalized to β-Actin, compared with mock-infected control, and expressed as mean ± SEM. (C, D) CD40L protein expression was determined by immunoblot analysis in whole brain lysates of RSA59 infected WT mice on days 5 and 10 p.i. Results were normalized to γ-Actin, compared with mock-infected control, and expressed as mean ± SEM. (E) Whole brain lysates of RSA59 infected WT and CD4$^{-/-}$ mice were subjected to qRT-PCR, and CD40L transcript abundance was compared between the two mouse strains on days 5 and 10 p.i. *Asterisk represents statistical significance calculated using unpaired Student's t-test, p<0.05 was considered significant. **p<0.01, ****p<0.0001. Data is represented from 3 independent biological experiments, N = 3.

critical regulator of cellular and humoral immune responses in viral immunopathogenesis [16–22]. It also has identified roles in several autoimmune diseases [23–28], including neuro-degenerative diseases such as MS [27, 29] and Alzheimer's [30]. T-helper cells in MS patients express CD40L; in fact, there is a positive correlation between the blood-brain-barrier break-down and CD40L(soluble CD40L) levels in MS patients' serum[31]. On the contrary, another study depicted decreased amounts of CD40L in the serum of MS patients, which further reduced upon exogenous IFN-β and glatiramer acetate administration[32]. While several reports in the EAE model of MS suggest that CD40 and CD40L interaction in the CNS is pathogenic [33–38], it is shown to be necessary for the protection in another demyelinating model of MS, Theiler's murine encephalitis virus (TMEV) induced neuroinflammation [39]. Notwithstanding the conflicting evidence, the CD40-CD40L dyad appears altered in MS. Therefore, we investigated the role of CD40-CD40L interaction as a differential etio-biology of MS.

Here, we show that RSA59 infection induces upregulation of CD40L in WT mice brains. However, CD4$^{-/-}$ mice exhibit extremely low expression of CD40L in the CNS, suggesting that CD4$^+$ T cells are the major cell population contributing to the expression of CD40L in the CNS post RSA59 infection. CD40L$^{-/-}$ mice show drastic weight loss, develop severe

neurological symptoms, and 50% of mice succumb to RSA59 at 50% of $LD_{50}$ dose by 10 days p.i., signifying that CD40L is required for mounting host-protective immunity. As a mechanism, microglial activation is reduced in CD40L$^{-/-}$ mice brains on days 5 and 10 p.i. CD40L$^{-/-}$ mice show decreased expansion of effector CD4$^+$ T cells in the cervical lymph node (CLN), correlating to their reduced numbers in the CNS. In contrast to the acute phase of neuroinflammation, chronic phase pathology was severely aggravated in CD40L$^{-/-}$ mice with persistently activated microglia/macrophages, axonopathy, and white matter loss or demyelination in the brain and spinal cord. Spinal cords of CD40L$^{-/-}$ showed persistent and profuse grey matter inflammation during the chronic infection, which was not observed in the wild-type mice. These results indicate the importance of CD40L in protection against virus-induced neuroinflammatory demyelination, demonstrating a sharp contrast with the autoimmune models of MS wherein CD40L plays a pathogenic role.

## Results

Age-matched 5 weeks old C57BL/6 wild-type (WT), CD4$^{-/-}$ and CD40L$^{-/-}$ male mice in the C57BL/6 background were inoculated intracranially with an isogenic enhanced green fluorescent protein (EGFP)-expressing RSA59 at 25000 PFUs for studies on day 5 p.i. and 10000 PFUs for days 7, 10 and 30 p.i. Detailed comparative histopathological studies combined with immunophenotyping by flow cytometry and mRNA and protein expression of different inflammatory mediators were performed at 5, 7, 10, and 30 days p.i. Mice were mock-infected (MI) with 0.75% PBS-BSA and maintained in parallel with the infected mice. All experimental mice were monitored daily for the development of clinical signs and symptoms.

### CD40L expression significantly increases upon RSA59 infection in WT mice

CD40L expression was examined in mock and RSA59 infected (25000 PFU) mouse brains on days 5 and 10 post-infection (p.i.). The relative fold change of mRNAs in the brains of infected WT mice was calculated compared to MI mice. Significant upregulation of CD40L mRNA expression was observed on day 10 p.i. in WT compared to the MI (Fig 1B) mice. CD40L protein showed significant upregulation on days 5 and 10 p.i. (Fig 1C and 1D). Evaluation of CD40L mRNA expression in infected WT and CD4$^{-/-}$ mice brains revealed significantly reduced expression of CD40L mRNA in infected CD4$^{-/-}$ compared to WT mice on days 5 and 10 p.i. (Fig 1E). Thus, the results implicate that CD4$^+$ T cells significantly express CD40L in the brain upon RSA59 infection.

### CD40L deficiency makes mice more susceptible to RSA59 infection

CD40L is primarily expressed by migrant CD4$^+$ T cells in the inflamed brain, and its receptor CD40 is abundantly expressed on microglia/macrophages[40]. CD4$^{-/-}$ mice with low levels of CD40L (Fig 1E) have also shown aggravated chronic progressive demyelination, accompanied by persistently activated microglia/macrophages in one of our previously published works [9]. Hence, the current study is focused on examining the molecular interaction between CD4$^+$ T cells and microglia/macrophage via CD40-CD40L co-stimulation in a virus-induced neuroinflammatory model of MS using CD40L$^{-/-}$ mice.

A comparative study was performed between WT and CD40L$^{-/-}$ mice at 25,000 PFUs of RSA59, and experimental mice were monitored daily for weight change and the development of clinical signs and symptoms. The mock infection did not show any differential phenotypic or histopathological features of interest for this study in WT and CD40L$^{-/-}$ mice (S1 Fig).

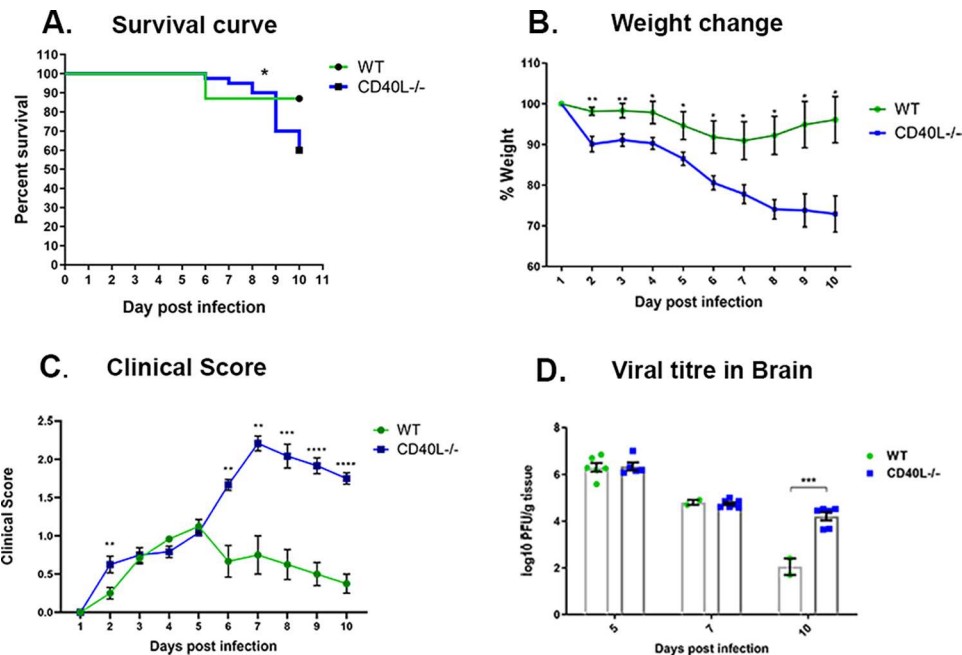

**Fig 2. CD40L deficiency makes mice more susceptible to RSA59 infection.** WT (N = 17) and CD40L[-/-] mice (N = 21) were infected with RSA59 (25000 PFUs) and monitored daily for (A) survival, (B) weight change, (C) development of the clinical disease, and (D) viral titers in brains. Clinical scores were assigned by a relative scale of 0–4 as described in Materials and Methods. Results were expressed as mean ± SEM. *Asterisk represents statistical significance calculated using unpaired Student's t-test, p<0.05 was considered as significant. Statistical significance for the survival curve was determined by Log-rank (Mantel-Cox) test. *p<0.05, **p<0.01, ***p<0.001, ****p<0.0001.

Upon RSA59 infection, CD40L[-/-] mice showed a significantly lower survival rate of 60% by day 10 p.i. in contrast with the survival rate of more than 86.9% in WT mice (Fig 2A). The decrease in the survival rate is associated with significant weight loss, clinical severity, high and persistent viral load. We observed that WT mice showed less than 10% weight loss, but CD40L[-/-] mice lost almost 30% of their original weight, which did not improve by day 10p.i. (Fig 2B). Most CD40L[-/-] mice displayed low clinical scores until day 5 p.i., then progressively increased to an average score of 2–2.5, indicated by partial/complete hind limb paralysis (Fig 2C). In contrast, WT mice never displayed clinical scores more than an average of 0.5 to 1, indicated by ruffled fur and the occasional presence of a hunchback phenotype (Fig 2C). The disease severity was correlated with viral titer in brains, as assessed by plaque assay. Viral load was equivalent in both mouse strains on days 5 p.i and 7 p.i. However, on day 10 p.i., CD40L[-/-] mice showed a significant persistence of viral titers compared to very low viral load just near the detection limit in WT mice (Fig 2D).

Disease severity also accompanied the persistence of systemic inflammation in the form of hepatitis (S2A, S2B and S2C Fig). However, no significant differences were observed in the liver viral titers on days 5 and 7 p.i. Infectious virus particles were below the detection limit in both WT and CD40L[-/-] mice by routine plaque assay on day 10 p.i. (S2D Fig).

Given the high susceptibility of CD40L[-/-] mice, as a result of persistent high viral loads in the brains and successive mortality, following inoculation with 50% of LD50 (25,000) of RSA59, subsequent experimental designs were lowered to 20% of LD50 (10,000 PFU) dose of RSA59 for studies beyond day 5 p.i.

## CD40L deficiency reduces encephalitis in the acute inflamed brain

H & E-stained brain sections from RSA59-infected CD40L[-/-] mice, like WT, showed acute encephalitis characterized by intraparenchymal perivascular lymphocytic cuff formation (black arrow) at the acute phase of infection (Fig 3A). Corresponding serial brain sections immunohistochemically stained with anti-Iba1 (Ionized binding adaptor protein-1, a pan-macrophage marker) confirmed Iba1[+] microglia/macrophage in the brain parenchyma (black arrowhead) and the perivascular region of both WT and CD40L[-/-] mice. The quantification of staining intensity revealed a significant reduction in the morphometric activation of Iba1[+] microglia/macrophages (Fig 3B).

## CD40L deficiency affects the migration and activation of monocytes/macrophages and microglia

We next performed flow cytometric analysis of the whole brain lysate to determine the absolute numbers of peripheral derived leukocytes and brain resident immune cells. Gating on CD45[hi] and CD45[lo] allowed the distinction between peripheral infiltrating leukocytes and brain resident immune cells, respectively[41]. Immunophenotyping revealed no significant differences between the mock-infected controls of WT and CD40L-/- mice. However, the overall CD45[hi] and CD45[lo] cell populations were markedly decreased in CD40L[-/-] mice (Fig 4A and 4B) compared to WT mice upon infection.

   We further examined the population of peripheral monocyte/macrophages (CD45[hi]CD11b[+] Ly6G[-]) and brain resident microglia (CD45[lo]CD11b[+]) at day 5 p.i. The analysis revealed a significant increase in the numbers of peripheral monocyte/macrophages as well as brain resident microglia in WT mice upon infection, but interestingly no significant alterations were observed in CD40L[-/-] mice, compared to their corresponding MI (Fig 4C and 4D). Thus, showing a significantly reduced pool of both peripheral derived monocyte/macrophages and brain resident microglia in the acute inflamed CD40L[-/-] mice brains compared to WT (Fig 4C and 4D). Further, activation of monocytes/macrophages/microglia was assessed by their expression of activation markers MHCII (major histocompatibility complex class II), CX3CR1 (C-X3-C chemokine receptor), and CD40. MHCII (Fig 4E and 4F), CX3CR1 (Fig 4G and 4H), and CD40 (Fig 4I and 4J) expression were reduced on both peripheral monocytes/macrophages and brain resident microglia in CD40L[-/-] mice as compared with WT mice. Together, these results suggest the importance of CD40L in the accumulation and activation of monocytes/macrophages and microglia in RSA59 induced acute neuroinflammation.

   No significant differences were observed in the lymphoid cell populations (CD4[+]T and CD8[+] T) between WT and CD40L[-/-] mouse brains on day 5 p.i. (S3 Fig). We also did not observe any significant differences in the transcripts of CCL5, CXCL5, CXCL9, and CXCL10 chemokines, responsible for the migration of lymphocytes into the CNS (S3 Fig).

## CD40L deficiency results in reduced activation of monocytes/macrophages and microglia during day 10 p.i.

We have previously shown that the accumulation of CD45[hi] and CD45[lo] immune cells infiltrates is maximum during days 7 and 10 post RSA59 infection[9]. This phase of the disease is considered the bridging phase between acute and chronic disease, wherein a notable shift is observed from innate immune responses towards the adaptive immune response (depicted in Fig 1A). We also observed that viral titer significantly declines on day 10 p.i. in the WT but remains almost 2 logs higher in CD40L[-/-] mice brains. We, therefore, performed flow

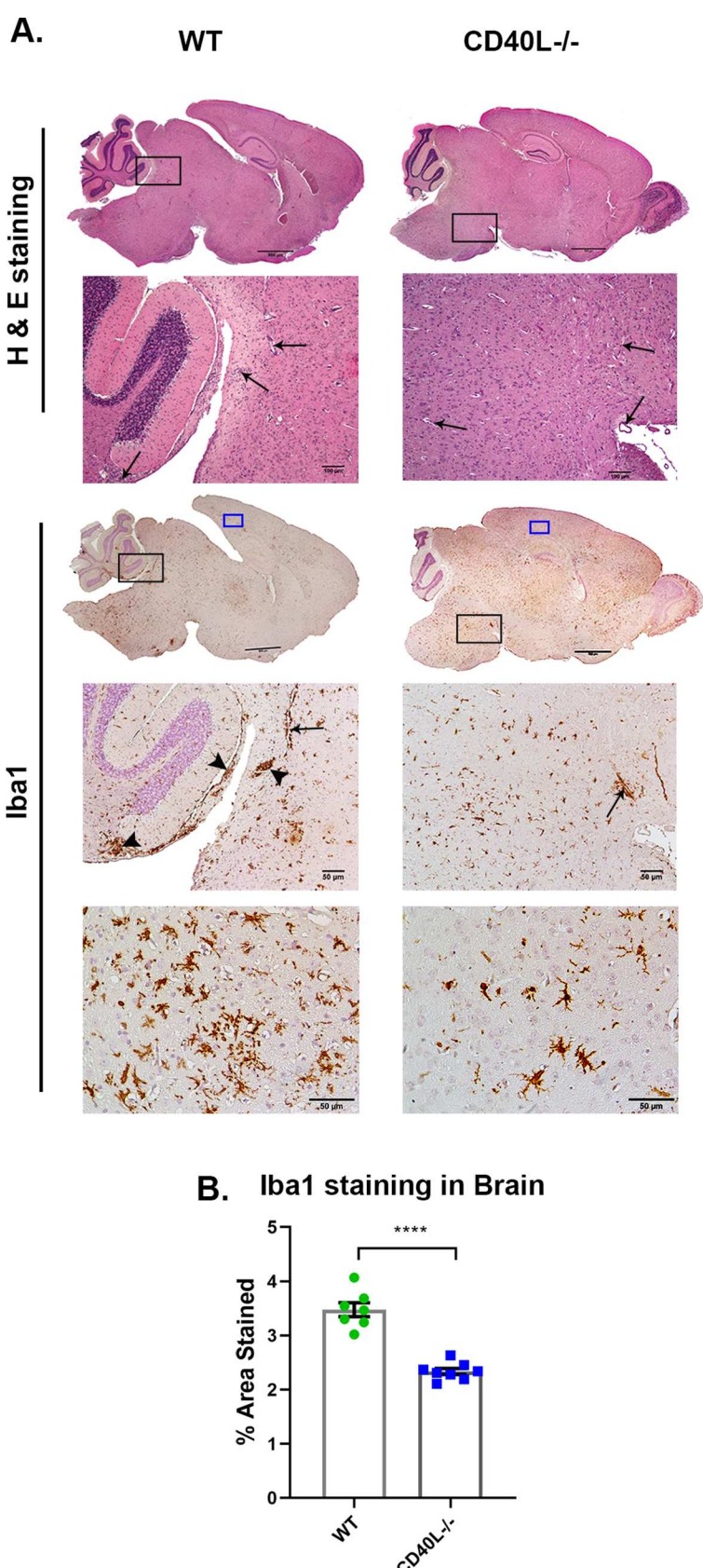

**B. Iba1 staining in Brain**

**Fig 3. CD40L deficiency results in reduced Iba1$^+$ microglial inflammation in the brain.** On day 5 p.i., sections of brains (A) from RSA59 infected (25000 PFUs) WT and CD40L$^{-/-}$ mice were stained with H&E and immunohistochemically for Iba1. The boxed areas are shown at higher magnification below the corresponding brain midsagittal sections. Black arrows in the zoomed sections mark characteristic perivascular cuffing, and black arrowheads mark microglial nodule formation. High magnification images corresponding to blue-lined rectangles from the cortical regions show the distinct morphology and distribution of Iba1+ cells. Scale bar, 500μm, 50μm. (B) Shows quantification of Iba1 staining in the brain. Results were expressed as mean ± SEM from 3–4 independent biological experiments (N = 4–5). *Asterisk represents statistical significance calculated using unpaired Student's t-test and Welch correction, p<0.05 was considered significant. ****p<0.0001.

cytometric analysis on day 10 p.i. to assess whether CD40L deficiency affects the immune cell infiltration in the brains during this phase.

Similar to results observed on day 5 p.i., CD40L$^{-/-}$ mice upon infection did not show a significant increase in the numbers of peripherally derived monocytes/macrophages (CD45$^{hi}$CD11b$^+$ Ly6G$^-$) and brain resident microglia (CD45$^{lo}$CD11b$^+$) compared to the respective MI (as shown in the graph in Fig 5B) on day 10 p.i. but WT mice showed increased numbers of both the cell populations. Therefore, compared to infected WT mice, the peripherally derived monocytes/macrophages and brain resident microglia were lower in the CD40L$^{-/-}$ mouse brains upon infection (Fig 5A and 5B). To assess whether the reduction in number was also associated with their activation state, both peripherally derived monocytes/macrophages and brain resident microglia were analyzed for their expression of activation markers MHCII, CX3CR1, and CD40. Results show that MHCII (Fig 5C and 5D) and CX3CR1 (Fig 5E and 5F) expressing cells were significantly reduced on day 10 p.i. in CD40L$^{-/-}$ mice compared to WT mice. Additionally, a marked reduction was also noted in CD40 receptor-expressing peripheral derived monocytes/macrophages and brain resident microglia in CD40L$^{-/-}$ mice on day 10 p.i. (Fig 5G and 5H).

Thus, flow cytometric analysis showed a critical role of CD40L in the accumulation and activation of monocytes/macrophages and microglia in the inflamed brain.

## CD40L deficiency impairs CD4$^+$ T cells infiltration/migration into the inflamed brain on day 10 p.i.

Since microglia/macrophage numbers were significantly diminished, we next examined the numbers of CD4$^+$ T lymphocytes in the brain. No differences were recorded in the recruitment of CD4$^+$ T cells in WT and CD40L$^{-/-}$ acute inflamed brain (day 5 p.i.). On day 10 p.i., no significant differences were observed between the WT and CD40L$^{-/-}$ MI mice, but significant increase was recorded in the CD4$^+$ T cell numbers upon infection in WT and CD40L$^{-/-}$ mice in comparison to their corresponding MI. Strikingly, upon comparing infected WT and CD40L$^{-/-}$ mice, we observed a significant decrease in their infiltration in CD40L$^{-/-}$ mouse brains on day 10 p.i. (S4A and S4B Fig). The expression of chemokine receptor CXCR3(C-X-C chemokine induces T cell migration)[42] was also assessed on CD4$^+$ T cells. MFI (median fluorescence intensity) of CXCR3 was comparable to WT in CD40L$^{-/-}$ mouse brains (S4C Fig). Thus, the impaired CD4$^+$ T cell trafficking observed in CD40L$^{-/-}$ mice was not due to their impairment of CXCR3 expression.

CD8$^+$ cytotoxic T cells showed no difference between WT and CD40L$^{-/-}$ mouse brains on day 10 p.i. (S5A and S5B Fig), similar to the acute phase.

## CD40L deficiency affects T cell priming and expansion in the CLN during days 7 and 10 p.i., resulting in their impaired infiltration into the brain

Cervical lymph nodes (CLN) are the initial T cell priming site during neurotropic MHV infection due to their anatomical linkage with CNS via the meningeal lymphatic system [43–45].

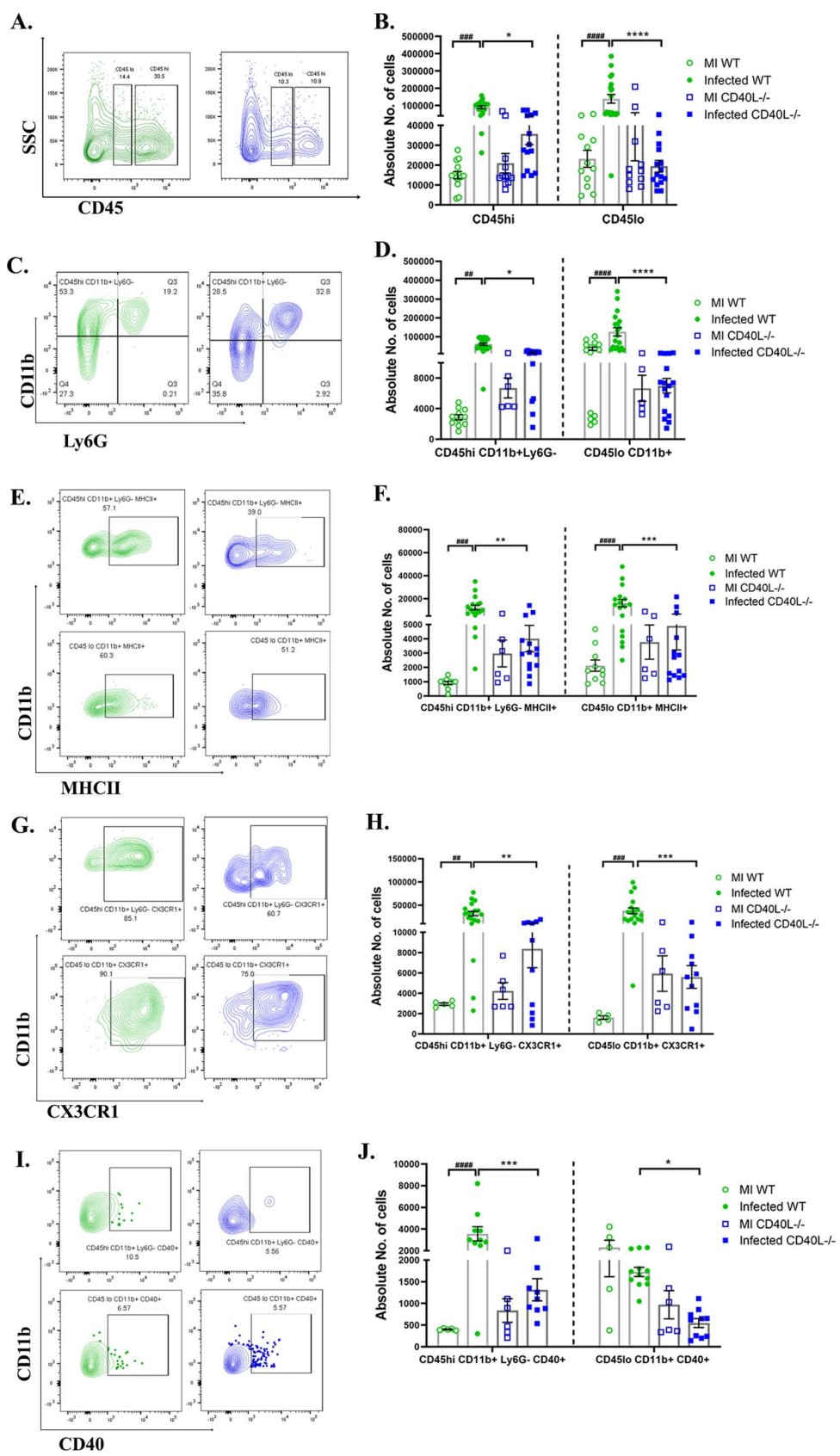

**Fig 4. CD40L deficiency causes impaired accumulation and activation of peripheral monocyte/macrophage and the brain resident microglia at the acute phase of inflammation.** On day 5 p.i., brains from MI and RSA59 infected (25000 PFUs) WT and CD40L$^{-/-}$ mice were harvested for flow cytometry analysis and stained for CD45, CD11b, MHCII, CX3CR1, and CD40. The green color denotes WT, and blue indicates CD40L$^{-/-}$ mice. (A) Representative flow cytometry contour plots showing percentages of overall CD45$^{hi}$ and CD45$^{lo}$ cell population after gating on singlets followed by live cells and absolute cell numbers from infected sets are represented in (B) comparing WT and CD40L$^{-/-}$ from MI and infected mice groups. Percentages of CD45 gated cells assessed for CD45$^{hi}$CD11b$^+$ (peripheral derived monocyte/macrophage) and CD45$^{lo}$CD11b$^+$ (brain resident microglia) from infected sets are presented in flow cytometry plots (C), and absolute numbers of MI and infected sets are graphically represented in (D) comparing WT and CD40L$^{-/-}$ groups. Representative flow cytometry contour plots indicate percentages of MHCII$^+$ (E), CX3CR1(G), and CD40 (I) expressing CD45$^{hi}$CD11b$^+$ and CD45$^{lo}$CD11b$^+$ cells comparing infected WT and CD40L$^{-/-}$ mice, and the absolute numbers of MI and infected sets are graphically represented in F, H, and J, respectively. Results were expressed as mean ± SEM from 3 independent biological experiments (N = 4). $^*$Asterisk (MI WT v/s infected WT) and #hash (infected WT v/s infected CD40L-/-) represents statistical significance calculated using unpaired Student's t-test and Welch correction, p<0.05 was considered significant. $^*$p<0.05, $^{**}$p<0.01, $^{***}$p<0.001, $^{****}$p<0.0001.

Therefore, to trace the cause of diminished CD4$^+$ T cell infiltration into the brain on day 10 p. i, we studied the CD4$^+$ T-cell priming and expansion and their activation in the CLNs of WT and CD40L$^{-/-}$ mice by flow cytometric analysis with a primary gating on CD3$^+$ before its infiltration in the CNS.

MI WT and CD40L$^{-/-}$ CLNs showed no significant differences in CD4$^+$ T cell populations. Upon infection, both WT and CD40L$^{-/-}$ mice show an increase in the numbers of CD4$^+$ T cells, CD4$^+$ CD44$^+$ effector CD4$^+$ CD25$^+$ Foxp3$^+$ Tregs compared to their respective MI. However, results comparing infected WT and CD40L$^{-/-}$ sets showed that the priming and expansion of CD4$^+$ helper T cells (Fig 6A and 6B) were significantly lower in CD40L$^{-/-}$ mice. The population of CD44 expressing CD4$^+$ (Fig 6C and 6D) effector T cells was significantly less in the CD40L$^{-/-}$ mice CLN compared to WT mice. CD44 expression on T cells is not only required for their effector functions, but it also regulates their infiltration into the target tissue [46]. Thus, the lower recruitment of CD4$^+$ T cells into the brain could be attributed to their impaired CD44 expression. Moreover, the number of CD4$^+$CD25$^+$Foxp3$^+$ regulatory T cells was also significantly lower in CD40L$^{-/-}$ mice (Fig 6E and 6F). These data signify that CD40L deficiency affects the priming and expansion of both subsets of CD4$^+$ T cell populations, helper T cells, and regulatory T cells in the CLN and their recruitment in the CNS upon RSA59 infection.

## CD40L deficiency exacerbates chronic phase (day 30 p.i.) inflammation, demyelination, axon degeneration, and grey matter inflammation in the brain and spinal cord

We further studied the consequence of impaired monocyte/macrophage and microglia activation and reduced CD4$^+$ T cell infiltration in the brain at the chronic phase of RSA59-induced neuroinflammation.

MI WT and CD40L$^{-/-}$ mice showed no observable inflammation in the brain and spinal cords on day 30 p.i. (S6 Fig). Comparative histopathological analyses of infected WT and CD40L$^{-/-}$ mouse brains on day 30 p.i. revealed higher inflammation in the brainstem, with vacuolar pathology characteristic of axon blebbing (Fig 7A) present in CD40L$^{-/-}$ mice. Corresponding areas (black box) showed that a significantly higher number of Iba1$^+$ activated and phagocytic microglia/macrophages were present in the vicinity of the vacuoles in CD40L$^{-/-}$ mice (Fig 7A). Iba1 staining intensity quantification revealed a marked increase in CD40L$^{-/-}$ mouse brains compared to WT mice (Fig 7B). Parallel sections also showed starkly faint density staining for myelin PLP (proteolipid protein), which indicated a lack of myelin in CD40L$^{-/-}$ mouse brains (Fig 7A). Axon blebbing, or the development of vacuoles due to the

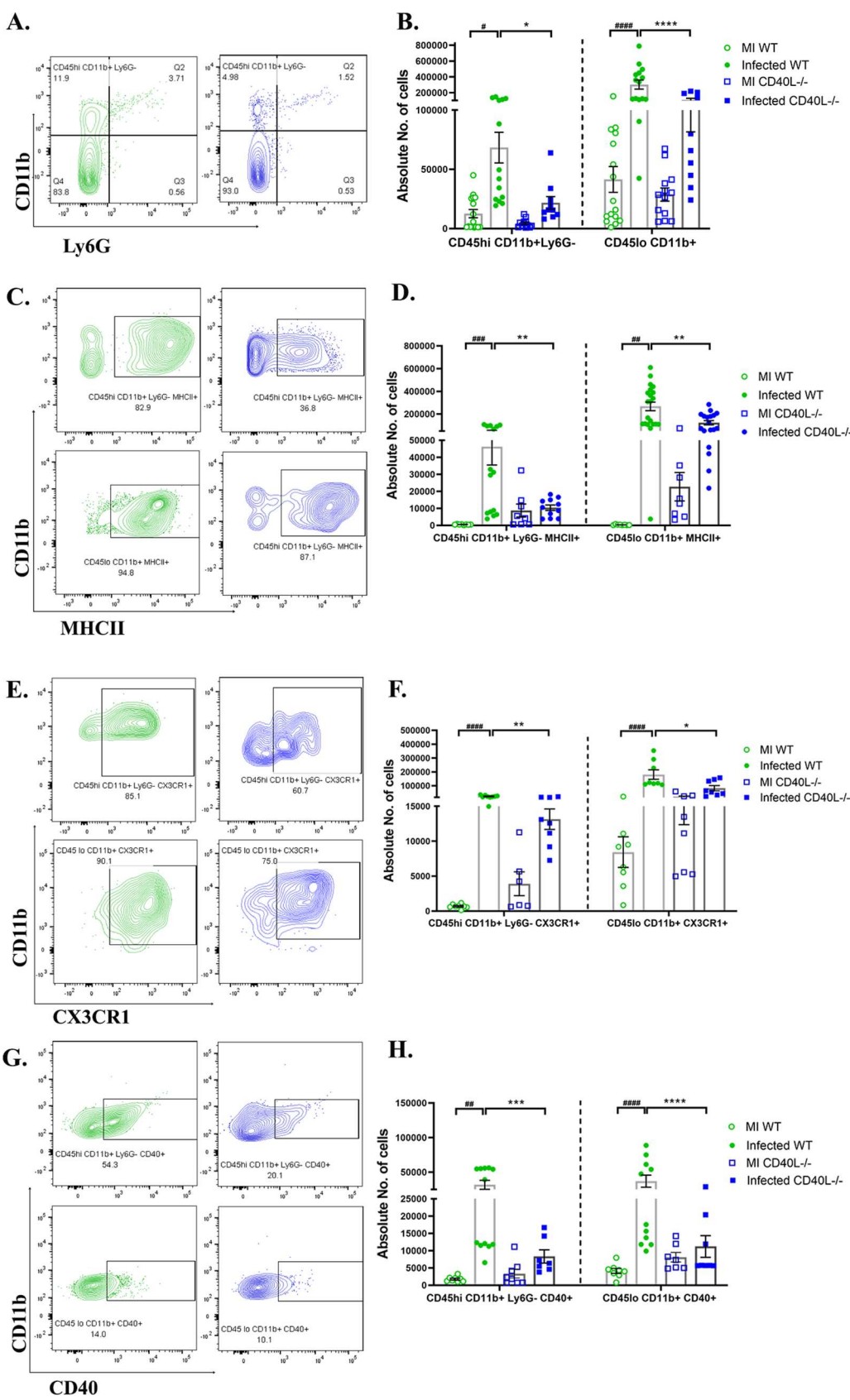

**Fig 5. CD40L deficiency results in reduced activation of monocytes/macrophages and microglia on day 10 p.i.** On day 10 p.i., cell suspensions from brains of MI and RSA59 infected (10000 PFUs) WT and CD40L$^{-/-}$ mice were analyzed as described in Fig 4 by flow cytometry following staining for CD45, CD11b, MHCII, CX3CR1, and CD40. The green color denotes WT, and blue indicates CD40L$^{-/-}$ mice. (A) Shows percentages of CD45$^{hi}$CD11b$^+$ (peripheral derived monocyte/macrophage) and CD45$^{lo}$CD11b$^+$ (brain resident microglia) from infected sets in representative contour plots. (B) shows quantification of absolute numbers comparing MI and infected WT and CD40L$^{-/-}$ groups. Representative contour plots from infected sets indicate the percentages of CD45$^{hi}$CD11b$^+$ and CD45$^{lo}$CD11b$^+$ cells expressing activation markers (C) MHCII, (E) CX3CR1, and (G) CD40 and the quantification are depicted in D, F, and H, respectively, comparing MI and infected WT and CD40L$^{-/-}$ groups. Results were expressed as mean ± SEM from 3 independent biological experiments (N = 4). *Asterisk (MI WT v/s infected WT) and #hash (infected WT v/s infected CD40L-/-) represents statistical significance calculated using unpaired Student's t-test and Welch correction, $p < 0.05$ was considered significant. *$p < 0.05$, **$p < 0.01$, ***$p < 0.001$, ****$p < 0.0001$.

inflammation of the neuropil, was evident in the magnified images of the white matter tracts stained with PLP, signifying denuded axons in CD40L$^{-/-}$ mice. Lack of myelin sheath as observed by PLP staining is associated with phagocytic microglia/macrophages, which have been previously established to trigger demyelination[4]. Therefore, we studied the transcript expression of selected markers known to be expressed by phagocytic microglia/macrophages [47–52]. Results show no difference in CD206 (Fig 7C) but significantly higher transcripts of P2Y6 (Fig 7D) and TREM-2 (Fig 7E) in CD40L$^{-/-}$ mice compared to WT mice. Conventionally, in RSA59 infected WT mice, the brain is immunologically silent on day 30 p.i., but CD40L$^{-/-}$ mouse brains were still in an inflamed state with the persistence of phagocytic microglia/macrophages, indicating the importance of CD40L in reinstating homeostasis.

RSA59 infection in 4 weeks old WT mice induces chronic phase (day 30 p.i.) demyelination, populated with microglia/macrophages (S7 Fig), whereas aging mice tend to develop a lower intensity demyelination. In this study, comparing 5-week-old mice, quite strikingly, RSA59 infected CD40L$^{-/-}$ mice exhibited aggravated inflammation in the dorsal/posterior columns, anterior horn, and/or lateral descending tracts, and ventrolateral white matter of the spinal cord upon H & E staining on day 30 p.i. They also displayed extensive vacuolar pathology with swollen axons throughout the white matter. In areas with grey matter and white matter inflammation in both WT and CD40L$^{-/-}$ mice (Fig 8A), corresponding serial sections stained for Iba1 showed the presence of a more significant number of microglia/macrophages in both white and grey matter regions of the CD40L$^{-/-}$ mice spinal cords compared to the WT mice (Fig 8A). In WT mice, inflammation was primarily resolved in the grey matter. Only a few activated/phagocytic microglia/macrophages were present in the white matter, corresponding to demyelination on LFB staining (blue arrow). In contrast, CD40L$^{-/-}$ mice displayed a large number of Iba1 immunoreactive activated cells in the grey matter, indicating the persistence of poliomyelitis until day 30 p.i., which had not been observed in previous studies (Fig 8A).

Additionally, Iba1$^+$ phagocytic cells were crowded in the white matter region (Fig 8A). Staining intensity quantification showed significantly higher expression of Iba1 in both grey and white matters of CD40L$^{-/-}$ spinal cords compared to WT spinal cords (Fig 8B). These microglia/macrophage-laden white matter regions corresponded to large demyelinating plaques upon LFB staining (blue arrows) (Fig 8B). Significantly higher demyelination was recorded in CD40L$^{-/-}$ mice spinal cords (Fig 8C). Corresponding demyelination regions further stained for PLP showed a significant loss in PLP in CD40L$^{-/-}$ mice spinal cord than WT spinal cord (Fig 8A).

RNA from day 30 p.i. spinal cords were subjected to qPCR analysis to assess microglial phagocytic markers' transcript levels. Results show higher mRNA expression of CD206 (Fig 8D), P2Y6 (Fig 8E), and TREM-2 (Fig 8F) in CD40L$^{-/-}$ spinal cords compared to WT spinal

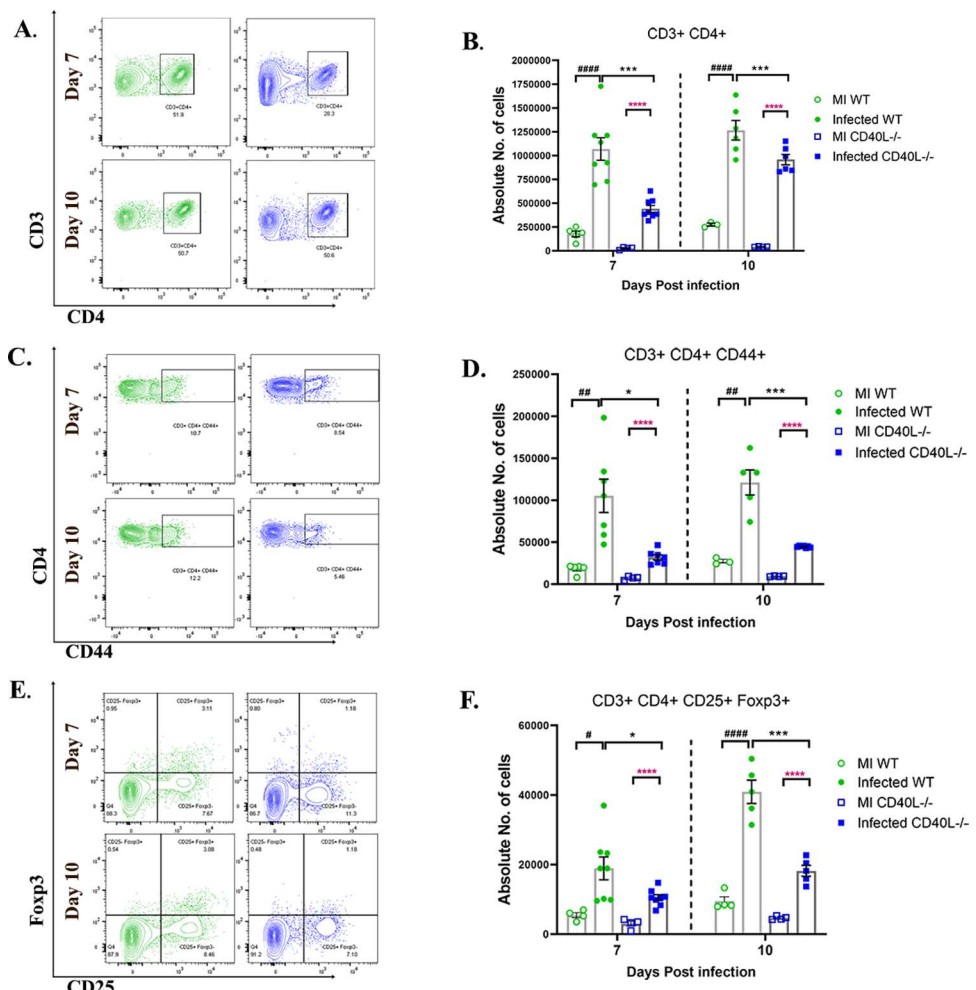

**Fig 6. CD40L deficiency affects T cell priming and expansion in the CLN on days 7 and 10 p.i.** On days 7 and 10 p. i., CLNs from MI and RSA59 infected (10000 PFUs) WT and CD40L[-/-] mice were harvested for flow cytometry analysis and stained for CD3, CD4, CD44, CD25, and Foxp3. The green color denotes WT, and blue indicates CD40L[-/-] mice. Primary gating was performed on singlets, followed by live cells and CD3. (A) Representative flow cytometry contour plots indicate the percentages of CD3[+]CD4[+] T helper cells from infected sets. (B) Graphical representation of absolute numbers of CD3[+]CD4[+] T helper cells comparing MI and infected WT and CD40L[-/-] groups across time points. (C) Representative flow cytometry contour plots indicating percentages of CD44[+] T helper cells from infected sets. (D) Shows the absolute numbers comparing MI and infected WT and CD40L[-/-] groups across time points. (E) CD3[+]CD4[+]CD25[+]Foxp3[+] T regs percentages from infected sets are represented in contour plots, and (F) absolute numbers of T regs are plotted comparing MI and infected WT and CD40L[-/-] groups across time points. Results were expressed as mean ± SEM from 3 independent biological experiments (N = 4). *Asterisk (black- MI WT v/s infected WT, pink- MI CD40L-/- v/s infected CD40L-/-) and #hash (infected WT v/s infected CD40L-/-) represents statistical significance calculated using unpaired Student's t-test and Welch correction, $p < 0.05$ was considered significant. *$p < 0.05$, **$p < 0.01$, ***$p < 0.001$, ****$p < 0.0001$.

cords. Thus, the results imply that the absence of CD40L results in an augmented microglia/ macrophage phagocytic phenotype and activation, which induces extensive myelinopathy.

## CD40L deficiency results in prolonged virus persistence in the CNS

Due to the severe chronic phase pathologies recorded in CD40L[-/-] mice, we next examined if the impaired microglia/macrophage and CD4[+] T cells response during the early disease phase also rendered mice incapable of clearing the virus. We quantified the viral nucleocapsid gene

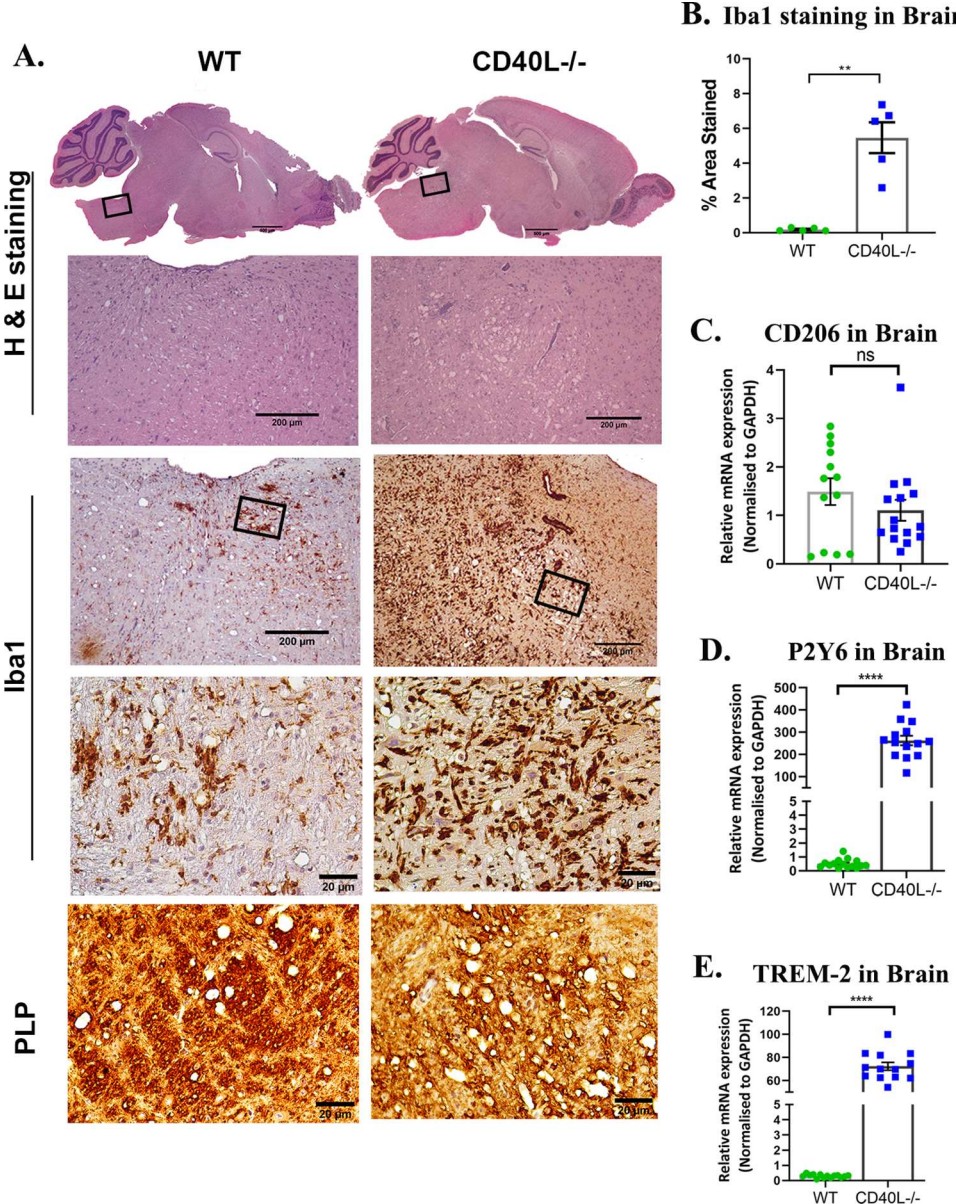

**Fig 7. CD40L deficiency results in severe chronic phase inflammation in RSA59 infected brains.** (A) Serial sagittal sections of brains from RSA59 infected (10000 PFUs) WT and CD40L⁻/⁻ mice were analyzed for inflammation on day 30 p.i. by H&E staining and immunohistochemically by microglia/macrophage marker Iba1 and myelin protein marker PLP. The black-lined rectangles in H & E-stained sections are shown at higher magnification below the corresponding brain midsagittal sections. The blue-line rectangle in the PLP sections is further magnified to show lateral white matter tracts. Scale bars 500μm, 200μm, 20 μm. (B) Quantification of Iba1 staining intensity. The relative gene expression of (C) CD206, (D) P2Y6, and (E) TREM-2 was analyzed using qRT-PCR of RNA from day 30 p.i. brains and compared between WT and CD40L⁻/⁻ mice. Results were expressed as mean ± SEM from 3 independent biological experiments (N = 5 for WT and N = 8 for CD40L⁻/⁻). *Asterisk represents statistical significance calculated using unpaired Student's t-test and Welch correction, p<0.05 was considered significant. **p<0.01, ****p<0.0001.

and immunohistochemically stained for viral nucleocapsid antigen distribution in the brain and spinal cord sections.

Quantitative real-time PCR results showed significantly higher transcript levels of the N gene in CD40L⁻/⁻ mice brains (Fig 9A) and spinal cords (Fig 9B) compared to WT on day 30 p.

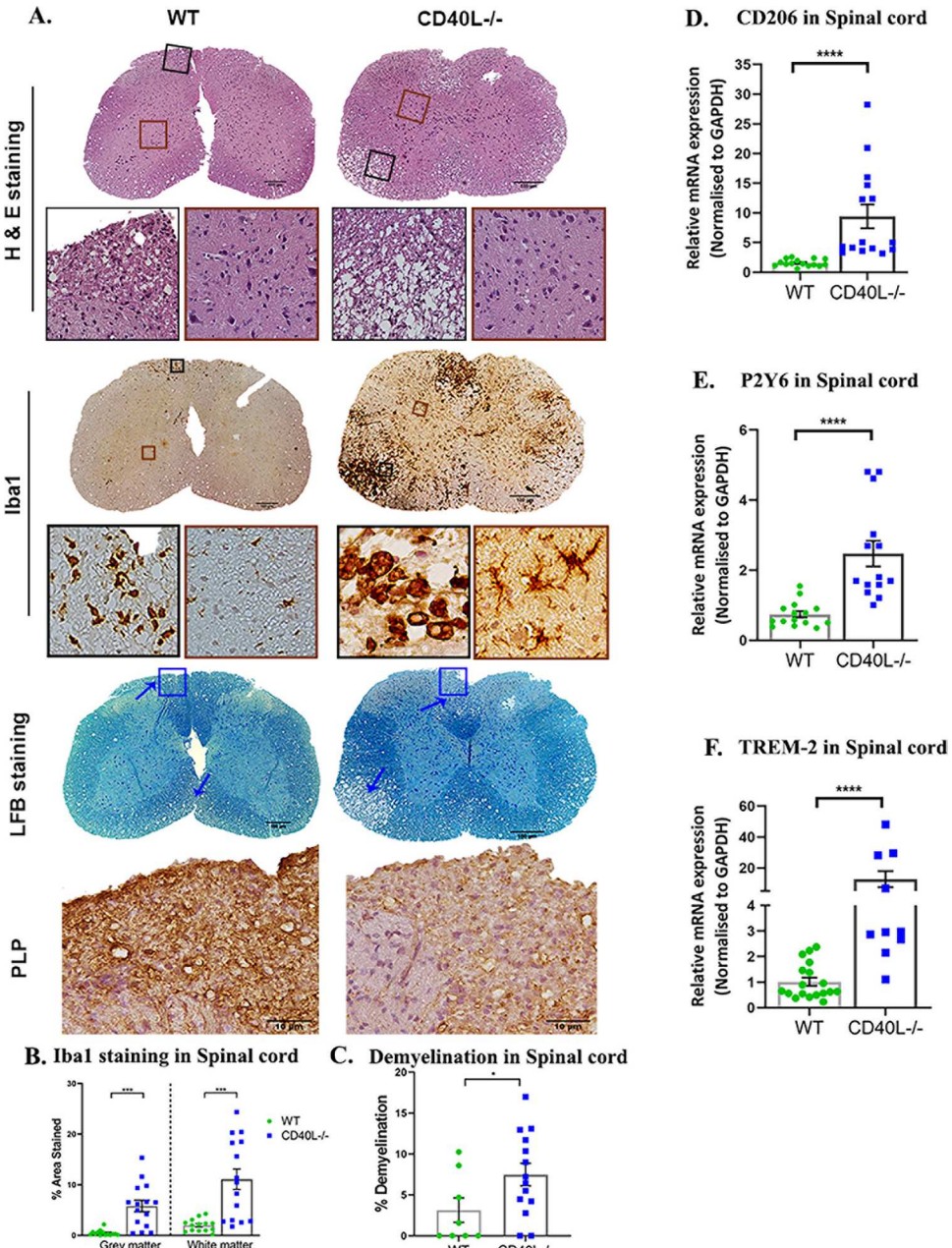

**Fig 8. CD40L deficiency exacerbates chronic phase inflammation, demyelination, axon degeneration, and grey matter inflammation in the RSA59 infected spinal cord.** (A) On day 30 p.i., cross-sections of RSA59 infected (10000 PFUs) WT and CD40L[-/-] mouse spinal cords were analyzed for the presence of inflammatory lesions by H&E, microglia/macrophages by Iba1, demyelination by LFB, and myelin protein marker PLP by immunohistochemistry. Black boxed areas represent higher magnification of white matter area, and brown boxed areas represent higher magnification of grey matter below the corresponding spinal cord cross-sections. Blue arrows mark demyelinating plaques on the LFB-stained sections. The blue lined box in the LFB stained sections corresponds to immunohistochemical staining for PLP protein below. Scale bar 100μm, 50μm, 10μm. Quantification of (B) Iba1 staining intensity in the grey and white matter and (C) percent area of demyelination. The relative transcript abundance of (D) CD206, (E) P2Y6, and (F) TREM-2 was analyzed in the infected spinal cords using qRT-PCR and compared between WT and CD40L[-/-] on day 30 p.i. Results were expressed as mean ± SEM from 3 independent biological experiments (N = 5 for WT and N = 8 for CD40L[-/-]). *Asterisk represents statistical significance calculated using unpaired Student's t-test and Welch correction, $p < 0.05$ was considered significant. *$p < 0.05$, ***$p < 0.001$, ****$p < 0.0001$.

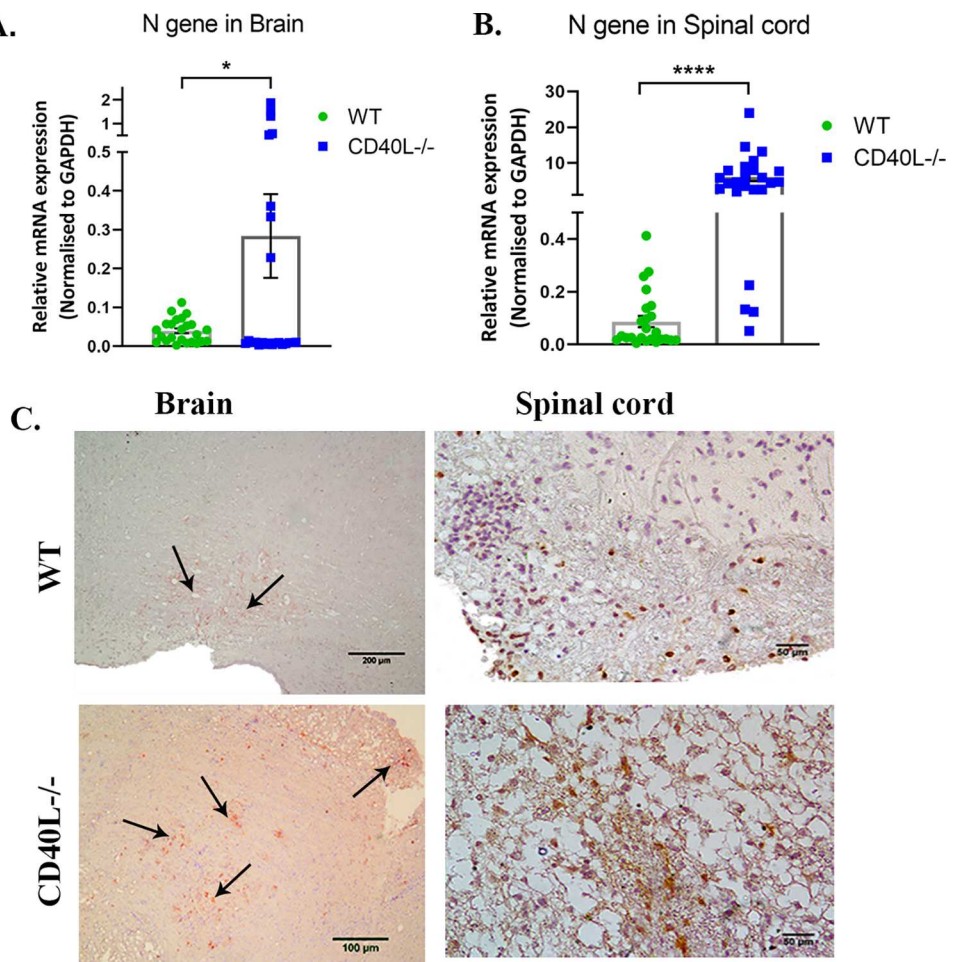

**Fig 9. CD40L deficiency results in prolonged virus persistence in the CNS.** (A). The relative abundance of transcripts corresponding to the viral N gene was compared using qRT-PCR in RSA59 infected (10000 PFUs) WT and CD40L[-/-] mice brains (A) and spinal cord (B). (C) Anti-N immunohistochemistry revealed the *in situ* distribution of viral N protein in the representative brain (brain stem) and spinal cord (dorsal column) anatomical regions in WT and CD40L[-/-] mice on day 30 p.i. Scale bars, brain-200μm, 100μm, and spinal cord- 50μm.

i. Also, viral nucleocapsid protein staining showed higher viral persistence in the CD40L[-/-] mice brain stem region and spinal cord dorsal columns than corresponding regions in the WT mice (Fig 9C).

Our results revealed prolonged viral persistence, sustained microglia/macrophage phagocytosis, and consequent demyelination in the CNS at day 30 p.i. This phenomenon is due to a mechanism of CD40L-dependent priming of CD4[+] T cells in the cervical lymph node at earlier time-points, impairment of which limits CD40L-dependent immune activation and infiltration of CD4[+] T cells in the CNS and their homeostatic interactions with microglia/macrophage.

## Discussion

RSA59 infection in WT mice causes acute phase meningoencephalomyelitis, which progresses to chronic phase demyelination[1]. Microglia/macrophage interaction with CD4[+] T cells is critical in the efficient transition from an innate immune response prevalent during the acute

phase to a chronic-adaptive inflammatory response observed during the chronic phase of RSA59 infection.

This study showed that RSA59 infection induces a significant upregulation of CD4$^+$ T cell activation marker and costimulant for CD40 on microglia/macrophage, CD40L in WT mice. Successive experiments using CD40L$^{-/-}$ mice highlighted a critical role of CD40L in RSA59-induced neuroinflammatory demyelination and axonal pathology. Our study highlights, (i) Absence of CD40L significantly enhanced RSA59 induced clinical severity and resulted in prolonged virus persistence in both brain and spinal cord; (ii) CD40L$^{-/-}$ mice displayed reduced activation of monocytes/macrophages and microglia in the brain, but no alteration in the lymphocyte infiltration during the acute phase of infection; (iii) CD40L$^{-/-}$ mice showed significantly lower infiltration of CD4$^+$ T cells in the brain on day 10 p.i. and displayed their dampened priming in the CLN during days 7 and 10 p.i. (iv) The absence of CD40L resulted in rare chronic poliomyelitis and significantly aggravated the demyelination pathology in the CNS, corresponding with profuse activation of phagocytic microglia/macrophages. Thus, results emphasize the crucial role of CD40L expression in modulating RSA59 induced neuroinflammatory demyelination.

Parallelly, comparing CD4$^{-/-}$ and WT mice, we have shown that CD40L is primarily expressed by CD4$^+$ T cells in the inflamed brain. A recent study showed that CD4$^+$ T cells acutely regulate the activation of microglia/macrophages in the CNS to reinstate neural homeostasis post neuroinflammation. Another study in Ifit2$^{-/-}$ mice showed reduced microglia/macrophage activation correlates to diminished CD4$^+$ T cells infiltration into the CNS[2]. Thus, in the current study, RSA59 infection in CD40L$^{-/-}$ mice has served as an efficient model to understand the molecular nexus between CD4$^+$ T cells and microglia/macrophages.

Significant weight loss in CD40L$^{-/-}$ mice accompanied with severe clinical symptoms and persistent hepatitis and high morbidity during days 8–10 p.i. when acute inflammation resolves, and adaptive immunity develops in WT mice[3, 53] signifies that CD40L$^{-/-}$ mice lack an efficient switch from acute innate to chronic adaptive protective immune responses. Interestingly, the viral titers in the liver of both WT and CD40L-/- mice decrease below the detection by day 10 p.i. but persist in the brains. This signifies the importance of CD40L in viral clearance, specifically in the brain.

Results of flow cytometric analysis further showed that lack of CD40-CD40L signaling in CD40L$^{-/-}$ mice significantly affected the numbers and activation of both CNS resident microglia and peripheral derived monocytes/macrophages during days 5 and 10 p.i. resulting in a dampened pro-inflammatory signature (CD40, CX3CR1, MHCII) on the microglia/macrophages. However, it is interesting to note that MI WT and CD40L$^{-/-}$ mice showed no differences in both these cell populations and their activation with respect to MHCII, CX3CR1, and CD40 expression. Upon infection, microglia/macrophages were significantly increased in number and activation only in WT mice. This further exemplified the importance of CD40L expression in RSA59 induced neuroinflammation. CD40 on microglia plays an essential role in both direct neuroprotection as well as modulation of immune responses that may lead to improved neurologic outcomes in CNS infections and related demyelinating disorders [54–56]. Likewise, CX3CR1 expression in the microglia has established roles in promoting neural homeostasis [2, 57, 58] apart from enabling inflammatory infiltration[59]. It was recently correlated with impaired microglial activation and diminished infiltration of peripheral lymphocytes in the CNS of Ifit2$^{-/-}$ mice [2]. The lower expression of CD40 and CX3CR1 in the absence of CD40L thus suggests a likely mechanism leading to reduced infiltration of peripheral CD4$^+$ T as shown in our study.

We show that CD4$^+$ T cell infiltration is also markedly reduced into the CNS of CD40L$^{-/-}$ mice during day 10 p.i. and not during the acute phase. Additionally, we also observed that the

CNS infiltrating CD4[+] T cells do not show impaired CXCR3, which is often regarded as the chemokine directing T lymphocytes' infiltration into the site of inflammation[42]. However, it was noteworthy that the expression of CD44 on the lymphocytes in the CLN was significantly reduced, which is the primary site of antigen-dependent maturation of lymphocytes. While CD44 has identified functions in T cell differentiation and activation, it is also strongly recognized as a promoter of migration of T lymphocytes to the site of infection[46]. Thus, the reduced infiltration of lymphocytes into the CNS may be attributed to the lower expression of CD44 and not CXCR3 chemokine. Low CD44 also resulted in a severe compromise of the effector functions of T lymphocytes. Our results also revealed substantially lower levels of Tregs in CD40L[-/-] mice. Tregs are known to limit acute neuroinflammation in the CNS by controlling anti-viral immune responses[60]. Studies have shown that the transfer of virus-specific Tregs significantly enhanced survival in MHV infected mice by limiting Th1 responses [44]. Possibly, the decreased involvement of Tregs and effector CD4[+] T cells, their reduced infiltration into the CNS, and impaired interaction with microglia/macrophages in CD40L[-/-] mice leads to persistent neuroinflammation as a result of viral persistence in the CNS, ultimately resulting in long-term neuronal damage. Our results showed that viral titer in brains was comparable in both mouse strains at the acute phase of infection when CD4[+] T cell function is not critical, as explained in the current study as well as previous studies[3, 9, 61]. CD4[+] T cell migration into the CNS is essential between the acute and chronic inflammatory phases to establish efficient adaptive immune responses against RSA59 and promote viral clearance by inducing anti-viral interferons.

Though we observed significant differences in CD4+ T cells response, we observed no significant differences in the infiltration of CD8[+] T cells in the CNS during days 5 and 10 p.i. as tested.

We have previously shown that the CD4[+] T cell deficiency causes the persistence of M2 phenotypic microglia/macrophages in the brain[9]. Our results using CD40L[-/-] mice show increased demyelination in the CNS and indicate persistent inflammation in the form of activated phagocytic microglia/macrophages. These microglia/macrophages present their characteristic amoeboid shape upon Iba1 staining surrounding white matter tracts of the brain and spinal cords along with significantly increased phagocytic markers' CD206, P2Y6, TREM-2 [52, 62, 63] at transcription levels. Interestingly, the morphology of Iba1[+] microglia/macrophages in the spinal cord grey matter and white matter is quite distinct. While the white matter harbors the characteristic, phagocytic, amoeboid microglia/macrophages[64] involved in the stripping of myelin sheaths leading to demyelination[4], grey matter shows numerous activated microglia/macrophages trying to reinstate homeostasis but are unable to achieve the said goal in the absence of homeostatic signals received from CD40L expressed on CD4[+] T cells and thus result in persistent grey matter inflammation in the form of poliomyelitis, which was also observed in CD4[-/-] mice previously only at the acute phase[9].

In conclusion, our results demonstrate that CD40L is critical for mounting a protective host immune response that is important in activating microglia/macrophages, eliminating the virus from the CNS, and playing an indispensable role in homeostasis. While many studies have revealed a pathogenic role of CD4[+] T cells and CD40L in other models of MS[27, 65, 66], we have successfully shown the protective role of CD4[+] T cells in our previous study and CD40L in the current study. Several years of work identified CD40-CD40L interaction as a strong therapeutic target in MS [40]. So much so that a neutralizing anti-CD40L mAbs (clone IDEC-131) showed promising results in inhibiting MS relapses during clinical trials, but the trial was halted due to severe thromboembolism in a patient with Crohn's disease[67]. The significant risk that remains associated with using anti-CD40L or anti-CD40 therapy is developing severe immune-suppressive side effects due to the crucial roles of this dyad in immune

activation shown in the current study. The CD40-CD40L interaction modulates various effector functions that serve as protective immune responses[68]. Further studies will be conducted in CD40 deficient mice to examine the role of CD40 individually in MHV infection and to identify and characterize the crucial cell types in MHV induced neuroinflammation. Together, these studies can help expand our knowledge intended to use CD40-CD40L as a potential target for MS treatment combined with current and novel immunomodulatory therapies.

## Materials and methods

### Ethics statement

CD40L$^{-/-}$ mice (Jackson Laboratory, B6.129S2-Cd40lg$^{tm1Imx}$/J, Stock no. 002770)[69] were obtained from the Animal Facility at National Centre of Cell Science Pune, India. All experimental procedures and animal care and use were strictly regulated and reviewed in accordance with good animal ethics approved by the Institutional Animal Care and Use Committee at the Indian Institute of Science Education and Research Kolkata (AUP no. IISERK/ IAEC/AP/ 2019/45). Experiments were performed following the guidelines of the Committee for the Purpose of Control and Supervision of Experiments on Animals (CPCSEA), India.

### Virus, inoculation of mice and experimental design

The hepatotropic and neurotropic (demyelinating) recombinant, EGFP expressing strain of MHV-A59, RSA59 was used in the study, as formerly described[70]. Five-week-old, MHV-free, wild-type (WT) C57BL/6 (B6) male mice (Jackson Laboratories), CD40L$^{-/-}$ mice (Jackson Laboratory, B6.129S2-Cd40lg$^{tm1Imx}$/J, Stock no. 002770) and CD4$^{-/-}$ (Jackson Laboratory, B6.129P2-*CD4$^{tm1Kik}$*/J, Stock no. 002663) were used for the study. The mice were inoculated intracranially in the right hemisphere with 25,000 (50% of LD50) and 10,000 PFUs of RSA59 strain. Likewise, mock-infected controls for WT, CD40L$^{-/-}$ and CD4$^{-/-}$ mice were inoculated with an endotoxin-free, filter-sterilized uninfected cell lysate (PBS+0.75% BSA) at an equivalent dilution, in a final volume of 20 μl. Mice were monitored daily post-infection (p.i.) for disease signs and symptoms. Clinical disease severity was graded regularly using the following scale: 0, no disease symptoms; 0.5, ruffled fur, possibly slower movement; 1.0, hunched back position with mild ataxia, possibly slower movement; 1.5, hunched back position with mild ataxia and hind-limb weakness, restricted movement; 2, ataxia, balance problem and/or partial paralysis but still able to move; 2.5, one leg completely paralyzed, motility issue but still able to move around with difficulties; 3, severe hunching/wasting/both hind limb paralysis and mobility is severely compromised; 3.5, severe distress, complete paralysis and moribund; 4, dead [71].

For histopathological and immunohistochemical analyses, mice were sacrificed at the acute infection phase, i.e., on day 5 (three mice per group), and chronic infection phase, i.e., day 30 (six to eight mice per group) post-infection. For RNA and protein studies, viral titer estimation and flow cytometry WT and CD40L$^{-/-}$ mice were sacrificed (three-four mice per group) on days 5, 7- and 10 post-infection.

### Estimation of viral replication

Approximately 120gm of brain tissues were homogenized in 2 ml of RPMI containing 25 mM HEPES (pH 7.2), using Tenbroeck tissue homogenizers. Following centrifugation at 450g for 10 min, cell pellets were taken for flow cytometry, and the supernatant was kept frozen at -80°C until titred for viral plaque estimation. RSA59 titers in the brain homogenate were

determined as described previously by standard plaque assay protocol on tight monolayers of L2 cells[70, 71].

## Histopathology and immunohistochemical analysis

Liver, brain, and spinal cord tissues were harvested and embedded in paraffin following trans-cardial perfusion with PBS. Tissues were post-fixed in 4% paraformaldehyde for 36–48 hours, following which tissues were processed in increasing concentrations of ethyl alcohol, xylene, and paraffin wax, embedded, and sectioned (5μm) for staining with Hematoxylin and Eosin for histopathologic analysis. Moreover, spinal cords were also stained with Luxol fast blue (LFB) to evaluate the demyelination as described previously[9].

Immunohistochemical staining of the brain and spinal cord tissue serial sections was performed as described previously, using avidin-biotin immunoperoxidase technique (Vector Laboratories) with 3, 3-diaminobenzidine as the substrate [13]. The following primary antibodies were used–1:500 dilution of anti-Iba-1 (Wako), 1:1 dilution of anti-PLP (Rat IgG, Gift from Judith B. Grinspan, Children's Hospital of Philadelphia, Philadelphia, PA), and 1:40 dilution of monoclonal antibody directed against the nucleocapsid protein (N) of MHV (monoclonal antibody clone 1-16-1 provided by Julian Leibowitz, Texas A&M University). Control slides from both WT and CD40L$^{-/-}$ mock-infected mice were stained in parallel.

## Quantification of histopathological sections

The functional scoring of the inflammatory lesions in the liver was characterized as the Hepatic activity index (HAI) based on the modified Knodell's HAI, commonly referred to as the Ishak system [9, 71].

Image analysis was performed using the basic densitometric thresholding application of Fiji (Image J, NIH Image, Scion Image) as described previously[72]. Briefly, image analysis for Iba1 sections was performed by capturing the images at the highest magnification (4X-for brain, 10X-for spinal cord) such that the entire section (i.e., scan area) can be visualized within a single frame. The RGB image was deconvoluted into three different colors to separate and subtract the DAB-specific staining from the background H&E staining. The perimeter of each brain and spinal cord tissue was digitally outlined, and the area was calculated in μm$^2$. A threshold value was fixed for each image to ensure that all antibody-marked cells were taken into consideration. The amount of Iba1 staining was termed as the '% area of staining.'

To determine the area of demyelination, LFB-stained spinal cord cross-sections from each mouse were chosen and analyzed using Fiji software (Image J, NIH Image, Scion Image) as described previously[72, 73]. Briefly, the total perimeter of the white matter regions in each cross-section was marked and calculated by adding together the dorsal, ventral, and anterior white matter areas in each section. Also, the total area of the demyelinated regions was outlined and collated for each section separately. The percentage of spinal cord demyelination per section per mouse was calculated.

## Protein isolation and western blot analysis

On average, 150mg of brain tissue was collected from euthanized mice following transcardial perfusion with 20ml PBS and flash-frozen in liquid nitrogen. Tissue was then lysed in 1 ml of RIPA buffer (0.1% SDS and 0.1% Triton X-100) with 1X complete mini protease inhibitor cocktail tablets (#11836153001; Roche). Brain tissues were homogenized by trituration and sonication. Tissue lysate was centrifuged at 13,500 RPM for 30 min at 4˚C, and the supernatant was collected as whole protein extract. Protein was quantified using Pierce BCA protein assay kit (Thermo Scientific, Rockford, IL, USA). Equal amounts of protein were resolved on

SDS-PAGE followed by transfer to polyvinylidene difluoride membranes (Millipore, Bedford, MA) using transfer buffer (25mM Tris, 192mM glycine, and 20% methanol). The membrane was subsequently blocked with 5% non-fat skimmed milk in TBST (Tris buffered saline containing 0.1% v/v Tween-20) for 1 hour at room temperature followed by incubation in primary anti-CD40L antibody (Sigma, 1:2000) (in blocking solution) overnight at 4°C. Membranes were then subjected to washes in TBST, followed by incubation with HRP-conjugated secondary IgG. Blots were washed in TBST, and immunoreactive bands were visualized using Super Signal Luminata Forte Western HRP Substrate (Millipore (India) Pvt Ltd). Densitometric analyses of non-saturated membranes were carried out using a Syngene G: Box chemidoc system and GENESys software.

## Flow cytometry analysis

Mice were transcardially perfused with PBS, and half brains were homogenized in 2 ml of RPMI containing 25 mM HEPES (pH 7.2), using Tenbroeck tissue homogenizers. Following centrifugation at 450g for 10 min, cell pellets were resuspended in RPMI containing 25 mM HEPES, adjusted to 30% Percoll (Sigma), and underlaid with 1 ml of 70% Percoll. Following centrifugation at 800 g for 30 minutes at 4°C, cells were recovered from the 30%-70% interface, washed with RPMI, and suspended in FACS buffer (0.5% bovine serum albumin in Dulbecco's PBS). Also, deep cervical lymph nodes were harvested, homogenized in 4 ml RPMI containing 25 mM HEPES (pH 7.2), passed through 70μm filters followed by 30μm filters to obtain single-cell suspensions. Following centrifugation at 450g for 10 min, cell pellets were resuspended in FACS buffer. Cells were counted using an automated cell counter (Invitrogen) to obtain the numbers of total leukocytes. 1 million cells were stained for flow cytometry.

Specific cell types in the brain and CLN were identified by staining with fluorochromes like fluorescein isothiocyanate (FITC), phycoerythrin [(PE), (PECy7)], peridinin chlorophyll protein [(PerCP), (PerCpCy5.5)], allophycocyanin [(APC), (APCCy7)] and violet excitable dyes [(V450), (V500)] conjugated MAb for 45 minutes at 4°C in FACS buffer. Expression of surface markers was characterized with MAb (all from BD Biosciences except where otherwise indicated) specific for CD45 (clone Ly-5), CD11b (clone M1/70), CD11c (clone HL3), CD40 (clone 3/23), CX3CR1 (clone SA011F11, Biolegend), MHCII (clone 2G9), Ly-6G (clone 1A8), CD3 (clone 145-2C11), CD4 (clone GK1.5), CD8 (clone 53–6.7), NK1.1 (clone PK136), CD44 (clone IM7), CD25 (clone PC61) and Foxp3 (clone G155-178). Intra-nuclear staining was performed for Foxp3 after fixation and permeabilization (BD 554714, 562574) as per the manufacturer's guidelines.

Samples were acquired on a BD FACSVerse flow cytometer (BD Biosciences) and analyzed on FlowJo 10 software (Treestar, Inc., Ashland, OR). [2, 44]. First, doublet exclusion using FSC-A and FSC-W was performed, and then cells were gated based on forward scatter (FSC), and side scatter (SSC) to focus on live cells. Further, the cells were analyzed to differentiate myeloid and lymphoid populations. Myeloid cells were gated from a primary gating on CD45, and an additional CD3 gating was applied for the lymphoid populations. Single colors and FMOs were used in all the experiments. Beads were gated based on their FSC/SSC pattern.

## Gene expression: RNA isolation, reverse transcription, and quantitative polymerase chain reaction

RNA was extracted from the brain or spinal cord tissues (flash-frozen) of RSA59 infected WT, CD4-/- and CD40L-/- and mock-infected mice using the Trizol isolation protocol following transcardial perfusion with DEPC treated PBS. The total RNA concentration was measured using a NanoDrop ND-2000 spectrophotometer. 1μg of RNA was used to prepare cDNA using

**Table 1. Sequence of Primers used in the study.**

| | Primer sequence (5'-3') | |
| --- | --- | --- |
| | **Forward Primer** | **Reverse Primer** |
| β-Actin | CTTCTACAATGAGCTGCGTGTG | GGTCTCAAACATGATCTGG |
| CD40L | GTGAGGAGATGAGAAGGCAA | CACTGTAGAACCGGATGCTGC |
| CCL5 | CCAATCTTGCAGTCGTGTTTGT | CATCTCCAAATAGTTGATGTATTCTTGAAC |
| CD206 | TGGAGGCTGATTACGAGCAGT | TTGGTTCACCGTAAGCCCAAT |
| CXCL5 | ACAGTGCCCTACGGTGGAAGT | GAGTGCATTCCGCTTAGCTT |
| CXCL9 | GGAACCCTAGTGATAAGGGAATGCA | TGAGGTCTTTGAGGGATTTGTAGTG |
| CXCL10 | GACGGTCCGCTGCAACTG | CTTCCCTATGGCCCTCATTCT |
| GAPDH | GCCCCTTCTGCCGATGC | CTTTCCAGAGGGGCCATCC |
| N-gene | GTTGCAAACAGCCAAGCG | GGGCGCAAACCTAGT |
| P2Y6 | TGCCAATCTACATGGCAGCA | CACGACTCCACACACTACCC |
| TREM-2 | CAGTGTCAGAGTCTCCGAGG | CACAGGATGAAACCTGCCTGG |

a High-Capacity cDNA Reverse Transcription Kit (Applied Biosystems). Quantitative Real-time PCR analysis was performed using DyNAmo Color Flash SYBR Green qPCR kit (Thermo Scientific) in a Step One plus Real-time PCR system (Thermo Fisher Scientific) under the following conditions: initial denaturation at 95°C for 7min, 40 cycles of 95°C for 10s, 60°C for 30s, melting curve analysis at 60°C for 30s. Reactions were performed in quadruplets. Relative quantitation was performed using the comparative threshold (ΔΔCt) method. mRNA expression levels of target genes in RSA59 infected WT, CD4$^{-/-}$, and CD40L$^{-/-}$ mice were normalized with GAPDH and expressed as relative fold change compared to their respective mock-infected controls. Primer sequences are mentioned in Table 1.

## Statistical analyses

The viral titer was calculated as plaque-forming units (PFU) based on the formula: no. of plaques X dilution factor/ml/gram of tissue and expressed as $\log_{10}$ PFU/gram of tissue. Quantitative RT-PCR data were presented as mean values ± SEM. Values were subjected to unpaired Student's t-test analysis and Welch correction for calculating the significance of differences between the means. Log-rank (Mantel Cox) test was used for calculating the statistical significance in mortality between groups. All statistical analyses were done using GraphPad Prism 8 (La Jolla, CA). A P-value of <0.05 was considered statistically significant.

## Supporting information

**S1 Fig. CD40 Ligand deficiency causes no significant pathology in mock-infected mice on day 5 p.i.** WT and CD40L$^{-/-}$ mice were infected with an uninfected cell lysate (PBS $^+$ 0.75% BSA). A. Five-micrometer-thick tissue sections were stained with H&E (liver, brain, and spinal cord) and anti-Iba1(brain and spinal cord) for routine histopathological studies. No significant inflammation was observed in WT and CD40L$^{-/-}$ mouse tissues. The data represents results from 3 independent biological replicates. Scale bars, 200μm.
(TIF)

**S2 Fig. CD40L deficiency causes severe hepatitis.** RSA59 infection is known to cause acute mild-moderate hepatitis. High mortality in CD40L$^{-/-}$ mice at a clinical score of less than 3.5 (moribund) and significantly high weight loss, thus suggested an investigation of systemic inflammation in the liver. (A) WT and CD40L$^{-/-}$ mice (N = 4–5 per time point) were subjected to histopathological analyses of liver tissues by H & E staining. Arrows show hepatic lesions,

manually enlarged lesions are depicted in insets. (B) The hepatic activity index was calculated according to Ishak's score, as described in Materials and Methods, and plotted. (C) The average numbers of hepatic lesions per section from each mouse were determined, and the combined results were plotted. (D) Viral titer was determined in the liver homogenates on day 5, 7 and 10 p.i. and plotted. Results were expressed as mean ± SEM. *Asterisk represents statistical significance calculated using unpaired Student's t-test, $p < 0.05$ was considered as significant. *$p < 0.05$, **$p < 0.01$. Scale bar 500 μm.
(TIF)

**S3 Fig. CD40L deficiency did not impair the accumulation of lymphoid cells in the CNS at the acute phase of inflammation.** On day 5 p.i. brains from WT and CD40L$^{-/-}$ mice infected with RSA59 were harvested for flow cytometry analysis and stained for CD45, CD4, and CD8. The green color denotes WT, and blue indicates CD40L$^{-/-}$ mice. Primary gating was performed on singlets followed by live cells and CD45. A, B. Representative flow cytometry plots indicating percentages and graphical representation of absolute numbers of CD45$^{hi}$CD4$^+$ and CD45$^{lo}$CD4$^+$ comparing WT and CD40L$^{-/-}$ groups. C, D. Representative flow cytometry plots showing the percentages and graphical representation of absolute numbers of CD45$^{hi}$CD8$^+$ and CD45$^{lo}$CD8$^+$ comparing WT and CD40L$^{-/-}$ groups. RSA59 infected brains of WT and CD40L$^{-/-}$ mice were subjected to qRT-PCR analyses of chemokines, E. CCL5, F. CXCL5, G. CXCL9, and H. CXCL10. Results were normalized to GAPDH, compared with mock⁻infected control, and expressed as mean ± SEM from 3 independent biological experiments (N = 3–5). Statistical significance was calculated using unpaired Student's t⁻test and Welch correction, $p < 0.05$ was considered significant.
(TIF)

**S4 Fig. CD40L deficiency significantly impairs CD4$^+$ T cell infiltration/migration into the inflamed brain on day 10 p.i.** On day 10 p.i., brain-derived cells from MI and RSA59 infected (10000 PFUs) WT and CD40L$^{-/-}$ mice were harvested for flow cytometry analysis and stained for CD3, CD45, CD4, and CXCR3. The green color denotes WT, and blue indicates CD40L$^{-/-}$ mice. Primary gating was performed on singlets, followed by live cells and CD3 and CD45. (A) Representative flow cytometry contour plots indicating percentages of CD45$^{hi}$CD3$^+$CD4$^+$ and histograms showing CD3$^+$CD45$^{hi}$CD4$^+$CXCR3$^+$ cells from infected sets. (B) Graphical representation of absolute numbers of CD3+CD45hi cells expressing CD4 and (C) Median Fluorescence Intensity of CXCR3 expression on CD3$^+$CD45$^{hi}$CD4$^+$ cells comparing MI and infected WT and CD40L-/- groups. Results were expressed as mean ± SEM from 3 independent biological experiments (N = 4). *Asterisk (MI WT v/s infected WT) and #hash (black-infected WT v/s infected CD40L-/-, pink- MI CD40L$^{-/-}$ v/s infected CD40L$^{-/-}$) represents statistical significance calculated using unpaired Student's t-test and Welch correction, $p < 0.05$ was considered significant. **$p < 0.01$, **$p < 0.01$, ****$pp < 0.0001$.
(TIF)

**S5 Fig. CD40L deficiency did not alter CD8$^+$ T cells infiltration into the mouse brain at the acute⁻adaptive bridging phase (Day 10 p.i.).** On day 10 p.i. brains from WT and CD40L$^{-/-}$ mice infected with RSA59 were harvested for flow cytometry analysis and stained for CD45, CD3 and CD8. The green color denotes WT, and blue indicates CD40L$^{-/-}$ mice. Primary gating was performed on singlets followed by live cells, CD45 and CD3. A, B. Representative Flow cytometry plots showing the percentages and graphs showing the absolute numbers of CD45$^{hi}$CD3$^+$CD8$^+$ comparing WT and CD40L$^{-/-}$ mice groups. Results were expressed as mean ± SEM from 3 independent biological experiments (N = 3). Statistical significance

calculated using unpaired Student's t test and Welch correction, p<0.05 was considered significant.
(TIF)

**S6 Fig. CD40 Ligand deficiency causes no significant pathology in mock-infected mice on day 30 p.i.** WT and CD40L$^{-/-}$ mice were infected with an uninfected cell lysate (PBS + 0.75% BSA). A. Five-micrometer-thick tissue sections were stained with H&E (brain, and spinal cord) and anti-Iba1(brain and spinal cord) for routine histopathological studies. No significant inflammation was observed in WT and CD40L$^{-/-}$ mouse tissues. The data represents results from 3 independent biological replicates. Scale bars, 50μm.
(TIF)

**S7 Fig. RSA59 infection induces demyelination in 4 weeks old WT mice.** 5μm thick serial cross sections from RSA59 infected WT mice spinal cords at day 30 p.i. were analyzed for A. presence of inflammatory lesions by H&E, B. inflammatory cells by anti-Iba1 (microglia/macrophage) by immunohistochemistry, and C. demyelination by LFB. Black arrows show Iba1$^{+}$ microglia/macrophages in B and corresponding demyelinating plaques in C. The data represents results from 2 independent experiments (N = 3). Scale bars, 50μm.
(TIF)

## Acknowledgments

We thank the Animal facility at the National Centre of Cell Science, Pune, India, for providing the CD40L$^{-/-}$ mice (Jackson Laboratory, B6.129S2-Cd40lg$^{tm1Imx}$/J, Stock no. 002770) used in the study. We also thank the IISER-Kolkata animal facility for providing the necessary support for all animal experiments

## Author Contributions

**Conceptualization:** Fareeha Saadi, Debanjana Chakravarty, Jayasri Das Sarma.

**Data curation:** Fareeha Saadi, Debanjana Chakravarty, Saurav Kumar, Mithila Kamble.

**Formal analysis:** Fareeha Saadi, Debanjana Chakravarty, Jayasri Das Sarma.

**Funding acquisition:** Jayasri Das Sarma.

**Investigation:** Fareeha Saadi, Debanjana Chakravarty, Jayasri Das Sarma.

**Methodology:** Fareeha Saadi, Debanjana Chakravarty, Saurav Kumar, Mithila Kamble.

**Project administration:** Jayasri Das Sarma.

**Resources:** Jayasri Das Sarma.

**Software:** Fareeha Saadi, Debanjana Chakravarty.

**Supervision:** Jayasri Das Sarma.

**Validation:** Jayasri Das Sarma.

**Visualization:** Fareeha Saadi, Debanjana Chakravarty, Jayasri Das Sarma.

**Writing – original draft:** Fareeha Saadi, Debanjana Chakravarty, Jayasri Das Sarma.

**Writing – review & editing:** Fareeha Saadi, Debanjana Chakravarty, Bhaskar Saha, Kenneth S. Shindler, Jayasri Das Sarma.

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
