## [Decision Letter · Decision Letter 0]

13 Jul 2021

Dear Dr Das Sarma,

Thank you very much for submitting your manuscript "CD40L protects against Mouse Hepatitis Virus-induced neuroinflammatory demyelination" for consideration at PLOS Pathogens. As with all papers reviewed by the journal, your manuscript was reviewed by members of the editorial board and by several independent reviewers. In light of the reviews (below this email), we would like to invite the resubmission of a significantly-revised version that takes into account the reviewers' comments. Please note that a revised version needs to include at least some suggested experiments;  the proposed transfer experiments for example, would make the study much stronger.

We cannot make any decision about publication until we have seen the revised manuscript and your response to the reviewers' comments. Your revised manuscript is also likely to be sent to reviewers for further evaluation.

Sincerely,

Matthias Johannes Schnell, PhD

Associate Editor

PLOS Pathogens

Volker Thiel

Section Editor

PLOS Pathogens

Kasturi Haldar

Editor-in-Chief

PLOS Pathogens

orcid.org/0000-0001-5065-158X

Michael Malim

Editor-in-Chief

PLOS Pathogens

orcid.org/0000-0002-7699-2064

Reviewer's Responses to Questions

**Part I - Summary**

Reviewer #1: in this study, Fareeha Saadi et al. show that mice deficient in CD40L are more susceptible than WT to neuroinflammatory demyelination induced by intracranial infection with mouse hepatitis virus. This is shown at 5, 10 and 30 days post infection after measuring a variety of parameters including virus loads, histopathology innate and adaptive cell infiltration in the brain, etc. Figure 2A requires more extended time of observation and statistical validation for the claim of difference in mortality. Most figures lack MI control (3, 4, 5, 6, 7, 8, 9, 10). and this is particularly important in Figure 7. In many micrographs, a greater magnification would help to see the differences. Some conclusions cannot be made. For example,CD4 T cell priming. More importantly, the data does not support that the increased disease in the C40L-/- mice is due to CD40L deficiency in CD4 T cells. This would require 1) Co-adaptive transfer experiment of CFSE (or similar stain) labeled WT and CD40L deficient CD4 T cells. 2) Reconstitute CD4 deficient mice with CD40L deficient or WT CD4 T cells. Some experiments

Reviewer #2: This is a very nicely done study by Saadi et al. examining the role of CD40L in the neurological damage following mouse hepatitis virus (RSA59) infection. The authors show that deficiency in CD40L results in increased neurological disease, which is associated with more severe pathology in early and late stages of disease as well as a decrease in CD4 T cells, microglia cells as well as monocyte/macrophages/neutrophil population (which was not sorted out). Long term damage was associated with prolonged virus in the brain. As the pathogenesis of virus-induced demyelination and damage to the CNS is still a strong area of interest and the authors study the interaction between the innate and adaptive immune response, this paper is of substantial interest to readers of PLOS Pathogens.

Reviewer #3: The Saadi et al. study describes the protective role of CD40L expressing CD4+T cells in reducing Mouse Hepatitis Virus-induced neuroinflammatory demyelination. The paper is well written and the experiments were well planned.

The authors performed a time course experiment assessing the brain and the cervical lymph node at different stages of MHV infection in wild-type and CD40L-/- mice, focussing on examining the molecular interaction between CD4+ T cells and microglia/macrophages. The lack of CD40L leads to increase in demyelination of axons, highlighting the protective role of this interaction.This result contrasts with the well-documented pathogenic role of the CD40-CD40L in an autoimmune disease model, thereby signifying this study.

**Part II – Major Issues: Key Experiments Required for Acceptance**

Reviewer #1: This paper is lacking in mechanism. It would be much improved with

1) Co-adoptive transfer of genetically marked and CFSE labeled WT and KO CD4 T cells and comparing their responses and migration in WT and CD40L -/- mice.

2) Show that the disease is recapitulated in CD4 deficient mice reconstituted with CD40L-/- mice while they are protected when given WT CD4 T cells.

In addition, the authors should see that all figures have MI control mice.

Reviewer #2: 1) One of the main questions of concern is what happens to the microglia population in the CNS of CD40L-/- mice. The number of resident brain microglia should not be changed dramatically unless the cells are undergoing apoptosis or are actively replicating. Do the numbers of microglia in WT mice increase compared to mock-infected animals? Alternatively, are the microglia in CD40L-/- mice dying, this could be determined by staining tissue sections for Iba1 and active Caspase 3.

2) the authors do not mention peripheral virus loads in the WT versus CD40L mice. It would be expected that they would be comparable based on the comparable loads in the brain at 5 and 7 dpi, but this should be addressed.

3) the experiment in Fig. 2 ends right when mice are developing clinical disease (day 10). The authors need to provide an explanation on why that time frame was not extended further. Was it due to the high level of weight loss in the CD40L-/- mice? Were any of the mice taken out past day 10?

Reviewer #3: None

**Part III – Minor Issues: Editorial and Data Presentation Modifications**

Reviewer #1: Line 46. Please define better the model. Introduce the virus, then explain the route of administration.

Lines 51-55. This is unclear. The model is about chronic infection but then talks about protective immunity. Please explain the model well and possible outcomes.

Line 110. Indicate route and pfu.

Figure 1A. Please describe in the introduction.

Line 132. From the data, it cannot be concluded that CD4 T cells are the primary population expressing CD40L. CD4 T cells could be involved in the recruitment of CD40L expressing cells.

Lane 145. Death of CD40L-/- mice does not seem to be less than 60%. Please indicate if there are significant differences or not in survival. Ideally, mice should be observed for a period of time, until death stabilizes.

Lane 171 should say Figure 3B.

Line 177. Indicate references demonstrating that CD45 intensity distinguishes cell origin.

Reviewer #2: 1) as the dose of inoculating virus changes between some experiments, it should be included in the figure legend.

2) lines 135-140 belong in the intro or discussion

3) the intro part of the abstract (lines 17-21) and should be rewritten

4) mock-infected controls in the flow data would be helpful to show increase/decrease in cell populations. Additionally, was a live/dead stain used to identify cells? What is the large negative cell population? Is that cellular debri?

5) the macrophage population was based on CD45 high, CD11b high - this population could also contain neutrophils. This should be mentioned.

6) magnification of axon blebbing (Fig. 8A) should be a higher resolution to observe

7) PLP staining was hard to discriminate from background stain in Fig. 8A

Reviewer #3: 1) Line 185: Please correct the figure number to figure 4 (CX3CR1)

PLOS authors have the option to publish the peer review history of their article (what does this mean?). If published, this will include your full peer review and any attached files.

Reviewer #1: No

Reviewer #2: No

Reviewer #3: No
---

## [Decision Letter · Decision Letter 1]

8 Oct 2021

Dear Dr Das Sarma,

Thank you very much for submitting your manuscript "CD40L protects against Mouse Hepatitis Virus-induced neuroinflammatory demyelination" for consideration at PLOS Pathogens. As with all papers reviewed by the journal, your manuscript was reviewed by members of the editorial board and by several independent reviewers. The reviewers appreciated the attention to an important topic. Based on the reviews, we are likely to accept this manuscript for publication, providing that you modify the manuscript according to the review recommendations.

Sincerely,

Matthias Johannes Schnell, PhD

Associate Editor

PLOS Pathogens

Volker Thiel

Section Editor

PLOS Pathogens

Kasturi Haldar

Editor-in-Chief

PLOS Pathogens

orcid.org/0000-0001-5065-158X

Michael Malim

Editor-in-Chief

PLOS Pathogens

orcid.org/0000-0002-7699-2064

Reviewer Comments (if any, and for reference):

Reviewer's Responses to Questions

**Part I - Summary**

Reviewer #1: I think the experiments provide an excellent characterization of the response of CD40L deficient mice to IC infection with MHV-A59. The authors infer but do not explore the mechanisms involved.

Reviewer #2: the authors have addressed all of my concerns. Nice job on the manuscript.

Reviewer #3: In this revised manuscript, Saadi et al. have addressed most of the concerns raised by the reviewer. Despite these additions, the revised manuscript is not that well written and has too many main figures.

**Part II – Major Issues: Key Experiments Required for Acceptance**

Reviewer #1: (No Response)

Reviewer #2: (No Response)

Reviewer #3: The manuscript contains too many main figures. Several figures can be moved to the supplementary section like Figs 2,4, 8,10 and 13.

**Part III – Minor Issues: Editorial and Data Presentation Modifications**

Reviewer #1: Please check the manuscript for typos and be sure you define all the acronyms the first time.

Reviewer #2: (No Response)

Reviewer #3: Line 232. On day ?

Line 290: The paragraph should mention that this study assesses day 30 p.i. in the beginning of the paragraph.

Line 371 word spacing

Line 372: RSA59 induced (n missing)

Fig 1 C: There is less actin in the Mock infected control suggesting a loading issue. This figure should be repeated or removed.

Fg 12: A-C: At what point was the spinal cord of 5 week old mice assessed ? day 30 p.i?

PLOS authors have the option to publish the peer review history of their article (what does this mean?). If published, this will include your full peer review and any attached files.

Reviewer #1: No

Reviewer #2: No

Reviewer #3: No

Figure Files:

Data Requirements:

Reproducibility:

References:

---

## [Editor Report · Decision Letter 2]

23 Oct 2021

Dear Dr Das Sarma,

We are pleased to inform you that your manuscript 'CD40L protects against Mouse Hepatitis Virus-induced neuroinflammatory demyelination' has been provisionally accepted for publication in PLOS Pathogens.

Best regards,

Matthias Johannes Schnell, PhD

Associate Editor

PLOS Pathogens

Volker Thiel

Section Editor

PLOS Pathogens

Kasturi Haldar

Editor-in-Chief

PLOS Pathogens

orcid.org/0000-0001-5065-158X

Michael Malim

Editor-in-Chief

PLOS Pathogens

orcid.org/0000-0002-7699-2064
---

## [Editor Report · Acceptance letter]

23 Nov 2021

Dear Dr Das Sarma,

We are delighted to inform you that your manuscript, " CD40L protects against Mouse Hepatitis Virus-induced neuroinflammatory demyelination ," has been formally accepted for publication in PLOS Pathogens.

Best regards,

Kasturi Haldar

Editor-in-Chief

PLOS Pathogens

orcid.org/0000-0001-5065-158X

Michael Malim

Editor-in-Chief

PLOS Pathogens

orcid.org/0000-0002-7699-2064